# AlphaResearch: Accelerating New Algorithm Discovery with Language Models

## ABSTRACT

Large language models have made significant progress in complex but easy-to-verify problems, yet they still struggle with discovering the unknown. In this paper, we present **AlphaResearch**, an autonomous research agent designed to discover new algorithms on open-ended problems by iteratively running the following steps: (1) propose new ideas (2) program to verify (3) optimize the research proposals. To synergize the feasibility and innovation of the discovery process, we construct a new reward environment by combining the execution-based verifiable reward and reward from simulated real-world peer review environment. We construct **AlphaResearchComp**, a new evaluation benchmark that includes an eight open-ended algorithmic problems competition, with each problem carefully curated and verified through executable pipelines, objective metrics, and reproducibility checks. AlphaResearch gets a 2/8 win rate in head-to-head comparison with human researchers. Notably, the algorithm discovered by AlphaResearch on the *"packing circles"* problem achieves the best-of-known performance, surpassing the results of human researchers and strong baselines from recent work (e.g., AlphaEvolve). Additionally, we conduct a comprehensive analysis of the benefits and remaining challenges of autonomous research agent, providing valuable insights for future research.

## 1 INTRODUCTION

Recent progress has shown that frontier LLMs like GPT-5 (OpenAI, 2025) and Gemini 2.5 (Comanici et al., 2025) could achieve expert-level performance in complex tasks such as mathematics (Trinh et al., 2024; Lin et al., 2025) and programming (Jimenez et al., 2024; Jain et al., 2025). While LLMs excel at processing and reasoning on problems that are within the boundary of existing human knowledge (Wang et al., 2024b; Phan et al., 2025), their capacity for independent discovery that pushes the boundaries of human knowledge still remains a question of paramount importance (Novikov et al., 2025). *Can these models create advanced knowledge or algorithms that surpass human researchers?*

Previous studies demonstrate that LLMs can generate novel ideas at a human expert level (Si et al., 2024; Wang et al., 2024a). However, the outcome evaluation of LLM-generated research ideas still struggles with biased verification methods (Ye et al., 2024) that constrain the exploration of out-of-boundary machine knowledge, such as LLM-as-a-judge (Lu et al., 2024), where misaligned LLMs are used to evaluate fresh ideas and inevitably favor solutions within existing knowledge boundaries. Furthermore, the ideation–execution gap (Si et al., 2025) between generating and executing new ideas also hinders models from producing advanced research outcomes. Moreover, prior attempts at autonomous algorithm discovery face a fundamental tension. Execution-based verification systems like AlphaEvolve Novikov et al. (2025) can rigorously validate whether code runs and meets constraints, but this verification alone might not be completely sufficient for discovery. For example, these systems could converge on technically correct but scientifically uninteresting or less impactful solutions—code that executes successfully yet offers no advancement over existing methods. Conversely, idea-generation systems evaluated purely by LLM judges can propose innovative concepts that prove computationally infeasible or violate problem constraints when implemented. The absence of real-world research environment rewards in execution-based agents and execution-

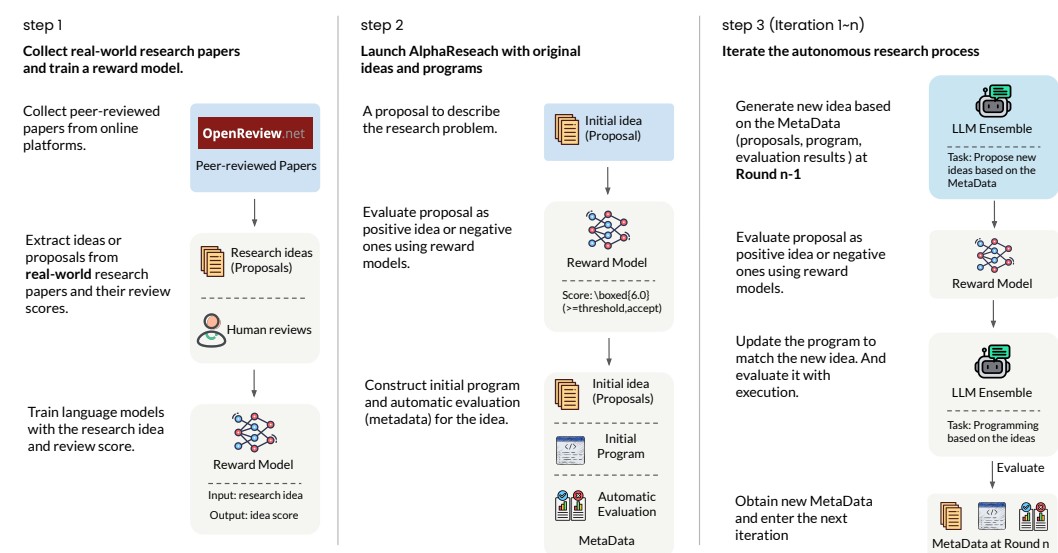

Figure 1: The launch of AlphaResearch contains two steps. (1) Train reward models with real-world peer-reviewed records. (2) Prepare initial research proposals, initial programs and evalution program. AlphaResearch will refine the research proposals and programs autonomously.

based reward in idea-generation systems renders the discovery of new knowledge and algorithms challenging for current autonomous research agents (Tian et al., 2024).

To combine the feasibility and innovation of the algorithm discovery process, we introduce **AlphaResearch**, an autonomous research agent that could discover new advanced algorithms with a suite of research skills including idea generation and code implementation that could interact with the environment. To synergize these research skills during the discovery process, we construct a novel dual research-based environment (Tian et al., 2024), where novel insights are forged by the simulated real-world peer-reviewed environment and execution-based verification. We use this dual environments to accelerate the discovery process because many research ideas can be evaluated before even implementing and executing on the idea. based on factors such as novelty, literature and the knowledge used. Specifically, we (1) train a reward model **AlphaResearch-RM-7B** with real-world peer-reviewed records, addressing the limitation of prior coding-only approaches that lack real-world research feedback, and use it to score the fresh ideas generated by LLMs; (2) construct an automatic program-based verifiable environment that executes these ideas with an interpreter. This dual environment facilitates a rigorous algorithm discovery process for autonomous research agents. As illustrated in Figure 1, AlphaResearch discovers new algorithms by iteratively running the following steps: (i) proposing new research ideas, (ii) verify the ideas in the dual research-based environment, and (iii) optimizing the proposals for higher reward from the environment. The synergy between an iterative real-world peer review environment and program-based verification empowers AlphaResearch to continuously explore novel research ideas and verify them via program execution. Once the generated optimal program surpasses current human-best achievements, these validated novel ideas could form feasible algorithms, thereby pushing the boundaries of human research forward.

To compare AlphaResearch with human researchers on novel algorithm discovery, we construct **AlphaResearchComp**, a simulated discovery **comp**etition between research agents and human researchers, by collecting 8 open-ended research problems and their best-of-human records (shown in Appendix I). Our results demonstrate that AlphaResearch surpasses human researchers on two problems but fails on the other six. The novel algorithms discovered by AlphaResearch not only surpass best-of-human performance but also significantly outperform the state-of-the-art results achieved by AlphaEvolve. Specifically, AlphaResearch optimizes the result of *"Packing Circles (n=32)"* problem to 2.939, where the goal is to pack $n$ disjoint circles inside a unit square so as to maximize the sum of their radii, surpassing the results of best-of-human and previous SoTA results achieved

---

**Algorithm 1** AlphaResearch

---

**Require:** initial idea $i_0$, initial program $p_0$, initial result $r_0$, model $\mathcal{A}$, evaluation program $\mathcal{E}(\cdot)$, maximum iteration rounds $n$,

1: $\tau_0 \leftarrow (i_0, p_0, r_0)$, $r_{best} = 0$            ▷ Initialization
2: **for** $k = 1$ to $n$ do **do**
3:     $(i_t, p_t, r_t) \sim \mathbb{P}(\cdot | \tau_{k-1})$            ▷ States Sampling
4:     $i_k \sim \mathbb{P}_{\mathcal{A}}(\cdot | i_t \oplus p_t \oplus r_t)$            ▷ New Idea Generation (Eq. 1)
5:     **if** $\mathcal{RM}(i_k) <$ threshold **then**
6:        **continue**            ▷ Reward Model for New Idea
7:     **end if**
8:     $p_k \sim \mathbb{P}_{\mathcal{A}}(\cdot | p_t \oplus i_k)$            ▷ Program Generation (Eq. 2)
9:     $r_k \leftarrow \mathcal{E}(p_k)$            ▷ Program-based Execution
10:    **if** $r_k > r_{best}$ **then**
11:       $(i_{best}, p_{best}, r_{best}) = (i_k, p_k, r_k)$
12:    **end if**
13:    $\tau_k \leftarrow \tau_{k-1} \oplus i_k \oplus p_k \oplus r_k$            ▷ Trajectory Update (Eq. 3)
14: **end for**
15: **return** $(i_{best}, p_{best}, r_{best})$

---

by AlphaEvolve (as shown in Appendix G). These entirely novel ideas and algorithms constitute the most advanced solutions currently present in the human knowledge base, demonstrating the feasibility of employing LLMs to advance the frontiers of human knowledge. The six failure modes in AlphaResearchComp demonstrate the challenges for the autonomous algorithm discovery with research agents. We analyze the benefits and remaining challenges of autonomous research agents for knowledge discovery, providing valuable insights for future work.

## 2 ALPHARESEARCH

### 2.1 OVERVIEW

AlphaResearch discovers out-of-boundary novel algorithms by continuously optimizing the research outcome from the dual reward that synergizes rigorous program verification and a simulated real-world peer review environment. As shown in Figure 1, given initial idea $i_0$ and program $p_0$, AlphaResearch runs the program $p_0$ with execution, producing $r_0$, which represents the initial overall rating. The triplet $(i_0, p_0, r_0)$ will be fed to AlphaResearch for subsequent processing, including newer idea generation, code implementation, and program-based execution. When reaching a point where execution output $r_n$ surpasses the previous rating, AlphaResearch will save the triplet $(i_{best}, p_{best}, r_{best})$ as the best record. We repeat the process until $r_{best}$ surpasses the best-of-human score, or the maximum round is reached. The resulting trajectory is denoted as $\tau = i_0 p_0 r_0 ... i_{n-1} p_{n-1} r_{n-1} i_n p_n r_n$, where $n$ is the total rounds.

### 2.2 ACTIONS

**New Idea Generation.** For each step $k$, AlphaResearch start with generating a new idea $i_k$ based on a sampled previous step $(i_t, p_t, r_t)$ from previous trajectory $\tau_{k-1} = i_0 p_0 r_0 ... i_{k-1} p_{k-1} r_{k-1}$. This process can be denoted as:

$$i_k \sim \mathbb{P}_{\mathcal{A}}(\cdot | i_t \oplus p_t \oplus r_t) \tag{1}$$

where $\oplus$ means concatenation, $t$ is the sampled step from trajectory $\tau_{i-1}$ and $\mathbb{P}_{\mathcal{A}}()$ indicates uniform sampling. We use a reward model to filter out high-quality ideas overall. If $\mathcal{RM}(i_n)$ outputs a negative score, we cease the subsequent actions in this round.

**Program-based Verification.** After obtain the fresh idea, AlphaResearch generates new program $p_k$ based on the previous implementation $p_t$ and new idea $i_k$ next:

$$p_k \sim \mathbb{P}_{\mathcal{A}}(\cdot | p_t \oplus i_k) \tag{2}$$

and yield the evaluation result $r_k$ by verifying $p_k$ with code executor $r_k \leftarrow \mathcal{E}(p_k)$. Then, we update the trajectory $\tau_k$ with the newly generated idea $i_k$, program $p_k$ and result $r_k$:

$$\tau_k \leftarrow \tau_{k-1} \oplus i_k \oplus p_k \oplus r_k \tag{3}$$

Table 1: Dataset for reward model training. We use the end of author-reviewer rebuttal period as the latest knowledge date.

| Split | Train | Test |
|---|---|---|
| | ICLR | ICLR |
| **Records** | | |
| **Range** | 2017∼2024 | 2025 |
| **Num** | 24,445 | 100 |
| **Start Date** | 2016-11 | 2024-10 |
| **End Date** | 2023-12 | 2024-12 |

Table 2: Evaluation results of RM. We use the more recent date between the model release date and the dataset cutoff as the latest date.

| Reward Model | Cutoff | Acc |
|---|---|---|
| Random (theoretical) | - | 50.0% |
| Human Annotator | - | 65.0% |
| GPT-5 (medium) | 2025-08 | 53.0% |
| Qwen2.5-7B-Instruct | 2024-09 | 37.0% |
| AlphaResearch-RM-7B | 2024-09 | 72.0% |

We repeat the above interaction process until $k$ reaches the maximum rounds $n$ and get the best result $(i_{best}, p_{best}, r_{best})$ as final output.

## 2.3 ENVIRONMENT

### 2.3.1 REWARD FROM REAL-WORLD RESEARCH RECORDS

Existing autonomous idea generation process suffers from a trade-off where highly novel research ideas may lack feasibility (Guo et al., 2025; Si et al., 2025). To address this gap and ensure the feasibility of idea candidates, we train a reward model with ideas from real-world peer-review information to simulate the real-world peer-review environment.

**Dataset for reward model.** To train our reward model (RM) to identify good ideas, we collect all ICLR peer review records from 2017 to 2024 as our training set. We sample a subset of ICLR 2025 records as a test set, where the dates of train and test are disjoint, which prevents knowledge contamination between the train and test split. We also select Qwen2.5-7B-Instruct as our base model, whose release date `2024-09` is earlier than the ICLR 2025 author-reviewer rebuttal period `2024-10`. For each record in the training dataset, we extract the abstract part as RM input and wrap the average peer-review overall ratings with `\boxed{}` as RM output. We fine-tune Qwen2.5-7B-Instruct with the RM pairs, yielding the AlphaResearch-RM-7B model.

**Can LLMs identify good ideas?** To simplify the RM evaluation, we binarize the RM output score according to the ICLR Reviewer Guide, where overall rating $> 5.5$ records are regarded as a positive score and $\leq 5.5$ records are negative. We compute the binary classification accuracy and evaluate three models (Deepseek-V3-0324, Qwen2.5-Coder-Instruct, and AlphaResearch-RM-7B) on the AlphaResearch-RM test set. Table 2 presents the evaluation results that eliminate the knowledge contamination, highlighting the following observations: (1) Both Deepseek-V3-0324 and Qwen2.5-7B-Instruct have lower than 50% accuracy when identifying the good ideas from ICLR 2025 records. (2) After fine-tuned with ideas from previous ICLR peer-review information, AlphaResearch-RM-7B demonstrates 72% binary classification accuracy on unseen ICLR 2025 ideas, significantly outperforming baseline models and human annotators. Based on these observations, we use the fine-tuned AlphaResearch-RM-7B as the final RM to simulate a real-world peer-review environment and filter out good ideas generated by AlphaResearch.

### 2.3.2 REWARD FROM PROGRAM-BASED EXECUTION

Inspired by AlphaEvolve (Novikov et al., 2025), we construct an automatic evaluation process with a code executor where each new program $p_k$ generated by AlphaResearch will be captured and evaluated. The evaluation program $\mathcal{E}(\cdot)$ includes two modules: (i) **Verification** module that validates whether $p_k$ conforms to the problem constraints. (ii) **Measurement** module that output the score $r_k$ of program performance. The program output $r_k$ will be injected into the idea generation prompt (if sampled), thereby participating in the optimization process for fresh ideas. These programs and results are stored in a candidate pool, where the primary goal is to optimally resurface previously explored ideas in future generations. The verifiable reward by code executor significantly simplifies the action spaces of AlphaResearch, thereby enhancing the efficiency of the discovery process.

Table 3: Problem overview in AlphaResearchComp. More information are shown at Appendix I.

| Problem | Human Best | Human Researcher |
|---|---|---|
| packing circles (n=26) | 2.634 | David Cantrell (2011) |
| packing circles (n=32) | 2.936 | Eckard Specht (2012) |
| minimizing max-min distance raio (d=2, n=16) | 12.89 | David Cantrell (2009) |
| third autocorrelation inequality | 1.4581 | Carlos Vinuesa (2009) |
| spherical code (n=30) | 0.67365 | Hardin & Sloane (1996, 2002) |
| autoconvolution peak minimization (upper bound) | 0.755 | Matolcsi–Vinuesa (2010) |
| littlewood polynomials (n=512) | 32 | Rudin–Shapiro (1959/1952) |
| MSTD (n=30) | 1.04 | Hegarty (2006/2007) |

## 3 ALPHARESEARCHCOMP

**Problems collection.** AlphaEvolve has not publicly disclosed all the test problems so far. To provide a transparent evaluation process, we curate AlphaResearchComp, a set of 8 frontier program-based research tasks spanning geometry, number theory, harmonic analysis, and combinatorial optimization. These problems were selected based on the following principles: AlphaResearchComp provides explicit, academically defined problem formulations, verification rules, and unified metrics (e.g., excel@best), enabling reproducible and controlled evaluation for open-ended discovery. This standardized pipeline design is essential for studying research agents.

- **Well-defined objectives.** Each task has a precise mathematical formulation with an objective function that admits rigorous automatic evaluation.

- **Known human-best baselines.** For every problem, we provide the best-known human result from the literature. These represent conjectured best-known values rather than proven optima, ensuring ample room for further improvement.

The curated problems are either inherited from prior work (e.g., AlphaEvolve) or collected from online repositories and domain experts. Each problem's baseline is supported by verifiable resources in the corresponding field. This design enables AlphaResearch to demonstrate both the *reproducibility* of established mathematical results and the *potential for discovery* beyond current human-best achievements. Detailed definitions, baseline values, and references for each problem are provided in the Appendix I.

**Initialization strategy.** After obtaining the research problems of AlphaResearchComp, we construct diverse initial states for each problem with the following strategies: (1) For the *"Packing Circles"* (n=26) and *"Packing Circles"* (n=32) problems, we initialize them with null programs ($r_0 = 0$) to simulate researches starting from scratch. (2) For the *"Littlewood Polynomials"* and *"MSTD (n=30)"* problems, we directly adopt the best-known solutions ($r_0 = r_{human}$) from human researchers to emulate improvements upon established methods. (3) For the remaining problems, we employ a moderate initialization strategy ($0 < r_0 < r_{human}$) to ensure sufficient room for the research agent to explore. This initialization strategy simulates a variety of real-world scenarios for the research agent, thereby facilitating a thorough evaluation process.

**Metrics.** For benchmarks like code generation with good verification techniques (e.g., unit tests), pass@k (Chen et al., 2021) is a metric denoting that at least one out of $k$ i.i.d. task trials is successful, which captures the ability of LLMs to solve easy-to-verified problems. For open-ended real-world algorithm discovery tasks, we propose a new metric - excel@best (excel at best), defined as the percentage excess on baseline (best of human level) results:

$$\text{excel@}best = \mathop{\mathbb{E}}_{\text{Problems}} \left[ \frac{(r_{best} - r_{human}) \cdot \mathbb{I}_d}{r_{human}} \right] \quad (4)$$

where $r_{human}$ indicates the results of human's best level. $\mathbb{I}_d$ indicates the optimization direction where $\mathbb{I}_d = 1$ represents that higher score is better and $\mathbb{I}_d = -1$ represents lower.

Table 4: Results on AlphaResearchComp. ↑ inidicates that higher score is better and ↓ for lower.

| Problem | Human | AlphaResearch | | Excel@best |
|---|---|---|---|---|
| | | *init* | *best* | |
| packing circles (n=26) ↑ | 2.634 | 0 | 2.636 | 0.32% |
| packing circles (n=32) ↑ | 2.936 | 0 | 2.939 | 0.10% |
| minimizing max-min distance ratio ↓ | 12.89 | 15.55 | 12.92 | -0.23% |
| third autocorrelation inequality ↓ | 1.458 | 35.746 | 1.546 | -6.03% |
| spherical code (d=3, n=30) ↑ | 0.6736 | 0.5130 | 0.6735 | -0.01% |
| autoconvolution peak minimization ↓ | 0.755 | 1.512 | 0.756 | -0.13% |
| littlewood polynomials (n=512) ↓ | 32 | 32 | 32 | 0 |
| MSTD (n=30) ↑ | 1.04 | 1.04 | 1.04 | 0 |

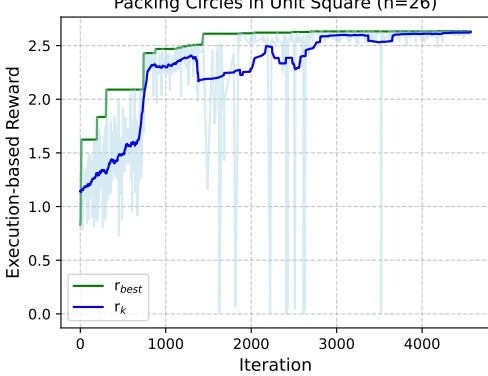 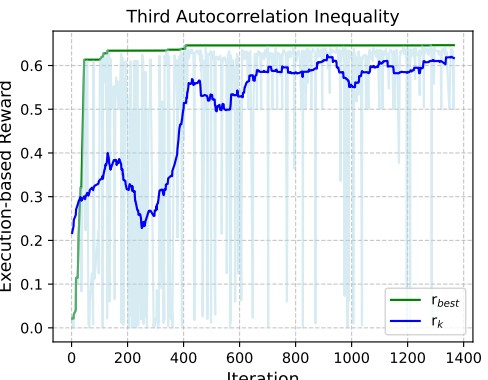

Figure 2: Execution-based reward of AlphaResearch on packing circles (n=26) problem (left) and third autocorrelation inequality problem (right).

## 4 EXPERIMENTS

### 4.1 SETUP

We select `o4-mini`, a strong but cost-efficient LLM as our research agent and run AlphaResearch on each problem to get the best algorithm. We perform supervised finetuning on `Qwen-2.5-7B-Instruct` (Yang et al., 2025) with the collected ICLR records, yielding AlphaResearch-RM-7B. We do not compute loss on paper information, only on the average rating scores within `\boxed{}`. For fine-tuning hyperparameters, we train our model with a learning rate of 1e-5 warmed up linearly for 100 steps. We train all the models in bfloat16 precision with Pytorch Fully Shard Data Parallel (FSDP) and set a global batch size to 128 for 2 epochs. All other settings not mentioned in this paper follow the default values of Huggingface Trainer [1].

### 4.2 RESULTS

**LLMs could sometimes discover new algorithms themselves.** Table 4 presents the results of AlphaResearchComp on 8 algorithms discovery problems. AlphaResearch achieved a 2/8 win rate (excel@best > 0) against human researchers, with one notable success: the algorithm discovered by AlphaResearch for *"Packing Circles"* problem reaches the best-of-known performance (2.636 for n=26, 2.939 for n=32), outperforming human researchers (2.634 for n=26, 2.936 for n=32) and AlphaEvolve (2.635 for n=26, 2.937 for n=32), where case (n = 32) is shown in Figure 10.

**LLMs can refine their research ideas autonomously.** AlphaResearch discovers advanced algorithms by iteratively proposing and verifying new research ideas. As shown in Table 2, 6/8 problems

---

[1] https://huggingface.co/docs/transformers/main_classes/trainer

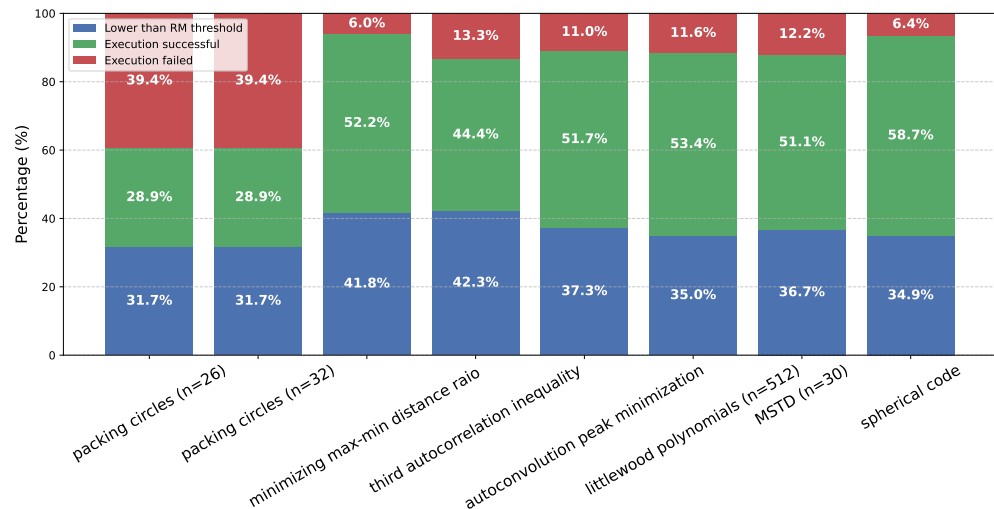

Figure 4: Reward overview during the discovery process. Each action in AlphaResearch will obtain 3 kinds of reward: (1) idea scrapping due to a lower RM score than the threshold (2) idea execution successes (3) idea execution fails.

demonstrate consistent improvement throughout the discovery process. Figure 2 presents two examples of the reward trend in AlphaResearch, where the execution-based reward initially grows rapidly, then slowly plateaus for optimal performance seeking. This improvement trend emphasizes the autonomous discovery ability of research agents.

**The discovery of superhuman algorithms remains challenging for LLMs.** As illustrated in Table 2, despite exhibiting continuous reward growth, AlphaResearch's performance still underperforms human researchers in 6 out of 8 problems. We initialize AlphaResearch with the best-known solution from human researchers on *"Littlewood polynomials"* and *"MSTD(n=30)"* problems, where AlphaResearch didn't show an increase in execution-based rewards. This indicates that current LLMs still struggle to consistently find better algorithms than human researchers.

### 4.3 ABLATIONS AND ANALYSIS

**Execution-only agent against AlphaResearch.**
To compare AlphaResearch with execution-only agents, we utilize AlphaResearch-RM-7B to evaluate the novelty of ideas generated by the execution-only agent and ideas produced by AlphaResearch. As illustrated in Figure 3, the ideas generated by AlphaResearch generally achieve higher scores than execution-only research agents. This illustrates that AlphaResearch tends to generate better ideas to get higher external rewards, thus facilitating a more effective research optimization process.

**Analysis of the discovery process.** We analyze the reward distribution in AlphaResearch discovery process. As shown in Figure 4, approximately 30%~40% of newly proposed ideas fall below the RM threshold and are thus discarded. The remaining ideas are executed, with the success rate of execution largely depending on the inherent characteristics of the problems. For example, the execution success rate on *"Packing Circles"* problem

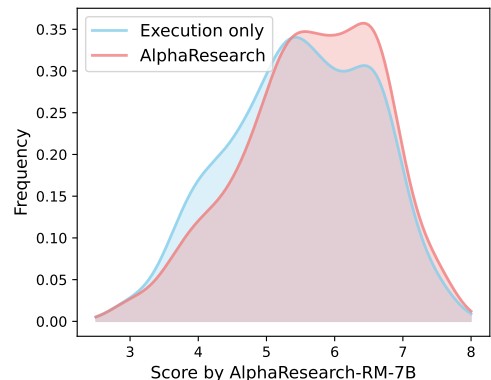

Figure 3: The idea comparison between the execution-only research agent and AlphaResearch, where AlphaResearch-RM-7B is used. This is done between the full distribution of all 1000 generated ideas from both agents without filtering.

is 28.9%, whereas it reaches 51.7% on the *"Third Autocorrelation Inequality"* problem. Figure 2 illustrates the execution-based rewards for these two examples in AlphaResearch. Despite the substantial variations in execution success rates, the execution-based rewards in both cases exhibit a consistent increasing trend. These findings demonstrate the interactions between LLM-based autonomous research agents and real-world environments.

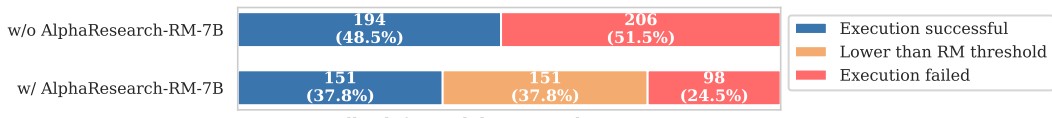

Figure 5: The impact of real-world peer review environment on execution results. AlphaResearch-RM-7B filters 151 bad ideas, where 108 ideas fail to execute and 43 are successful.

**The impact of real-world peer-review environment.**    To assess the effectiveness of reward from a simulated real-world peer-view environment, we ablate AlphaResearch-RM-7B at the first 400 iterations on *"Packing Circles"* problem. Figure 5 presents the execution results of w/ and w/o AlphaReasearch-RM-7B during the discovery process. Compared to the baseline without RM, AlphaResearch-RM-7B successfully filtered 151 ideas below the threshold. This process yielded 108 correct rejections of execution failures while making 43 erroneous rejections of viable ideas. AlphaResearch attained an accuracy of 71.5% (108/151), a result that aligns closely with its performance on the AlphaResearch-RM test set, as shown in Table 2 This outcome effectively demonstrates the model's generalization capabilities and the efficacy of incorporating feedback from a simulated real-world peer-review environment.

## 4.4  CASE STUDY

We select the successful example from AlphaResearch to better understand the discovery process. We'll consider the problem *"Packing Circles"* where the goal is to pack $n$ disjoint circles inside a unit square so as to maximize the sum of their radii, shown in Figure 6. We first initialize AlphaResearch with an original research proposal and a related program that returns a list of circles $(x, y, r)$ as output, as shown in Appendix I.4. The verification program first employs `verify_circles` function to check if the outputs of the initial program meet the problem constraints (e.g., all circles are inside a unit square) and `evaluate` function to output the sum of their radii. The metadata, including: (1) research ideas, (2) programs, (3) execution results, are subsequently preserved as candidates which represent the end of one step. At the next step, AlphaResearch will sample from the candidate pool and generate a new idea to improve the research proposals from the sampled metadata. After generating the new research ideas, AlphaResearch will further generate a patch to modify the existing program if the idea obtains a positive score from AlphaResearch-RM. The new program is then evaluated by the same verification program, thereby generating new metadata. We select the best program and idea as the final solution of AlphaResearch in this iterative process.

## 5  RELATED WORK

**LLMs for New Ideas.**    Several recent works explored methods to improve research idea generation, such as iterative novelty refinement (Wang et al., 2024a; Baek et al., 2024). These works focus on improving the research idea over vanilla prompting but critically miss an effective verification method. To promote more reliable AI-generated research ideas, many studies have proposed solutions from different perspectives, such as comparisons with any human expert (Si et al., 2024), using LLMs for executing experiments by generating code with human-curated research problems (Huang et al., 2024; Tian et al., 2024), and executing LLM-generated research ideas with LLM-generated programs (Li et al., 2024; Lu et al., 2024; Aygün et al., 2025). These works either use automatic program evaluation or a misaligned LLM evaluator method, which presents a challenge for their scalability to real-world advanced algorithm discovery. Our AlphaResearch presents a more feasible direction by combining program execution with RM training from real-world peer-reviewed research records.

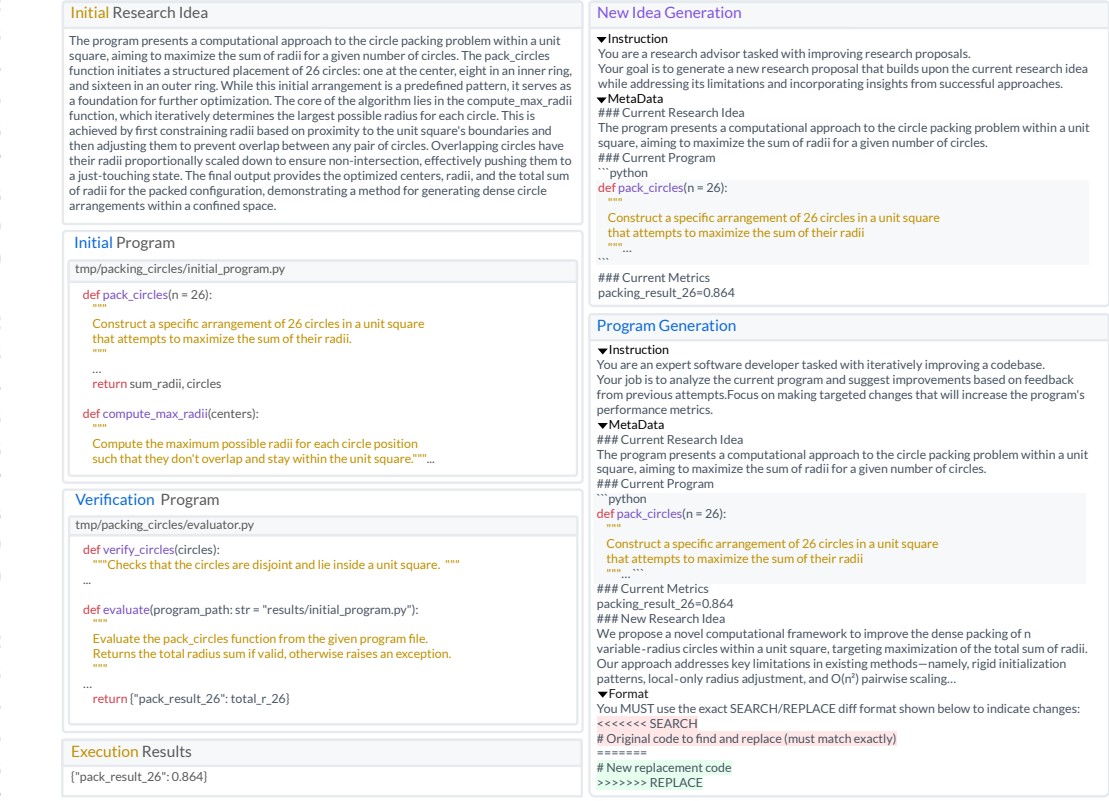

Figure 6: We show an example of a formatted task of AlphaResearch.

**LLMs for Code Generation.** In autonomous research agents, code generation serves as a fundamental step. Previous models (Guo et al., 2024; Yu et al., 2023; Hui et al., 2024) and benchmarks (Chen et al., 2021; Yu et al., 2025) for code generation are in a longstanding pursuit of synthesizing code from natural language descriptions. SWE-Bench (Jimenez et al., 2024), PaperBench Starace et al. (2025), MLE-Bench Chan et al. (2024) introduces the problems in real-world agentic coding. Many studies on SWE-Bench have greatly contributed to the emergence of coding agents like SWE-Agent (Yang et al., 2024) and OpenHands (Wang et al., 2025). These agent frameworks greatly facilitate the training of agentic LLMs like Kimi-K2 (Team et al., 2025) and GLM-4.5 (Zeng et al., 2025). The surge of these models on SWE-Bench underscores a critical need to reassess the future directions of coding agent research. Our AlphaResearchComp benchmark shows that testing LLMs on open-ended research for algorithm discovery is a promising direction to adapt language models to real-world tasks.

# 6 CONCLUSION

We present AlphaResearch, an autonomous research agent that synergistically combines new idea generation with program-based verification for novel algorithm discovery. Our approach demonstrates the potential of employing LLM to discover unexplored research areas, enabling language models to effectively tackle complex open-ended tasks. We construct AlphaResearchComp, including 8 open-ended algorithmic problems, where AlphaResearch outperforms human researchers in 2/8 algorithmic problems but lags behind in the remaining 6 problems.which demonstrates the remaining challenges of autonomous algorithm discovery for future research.

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

# APPENDIX CONTENTS

# A    COMPARISON WITH OPENEVOLVE

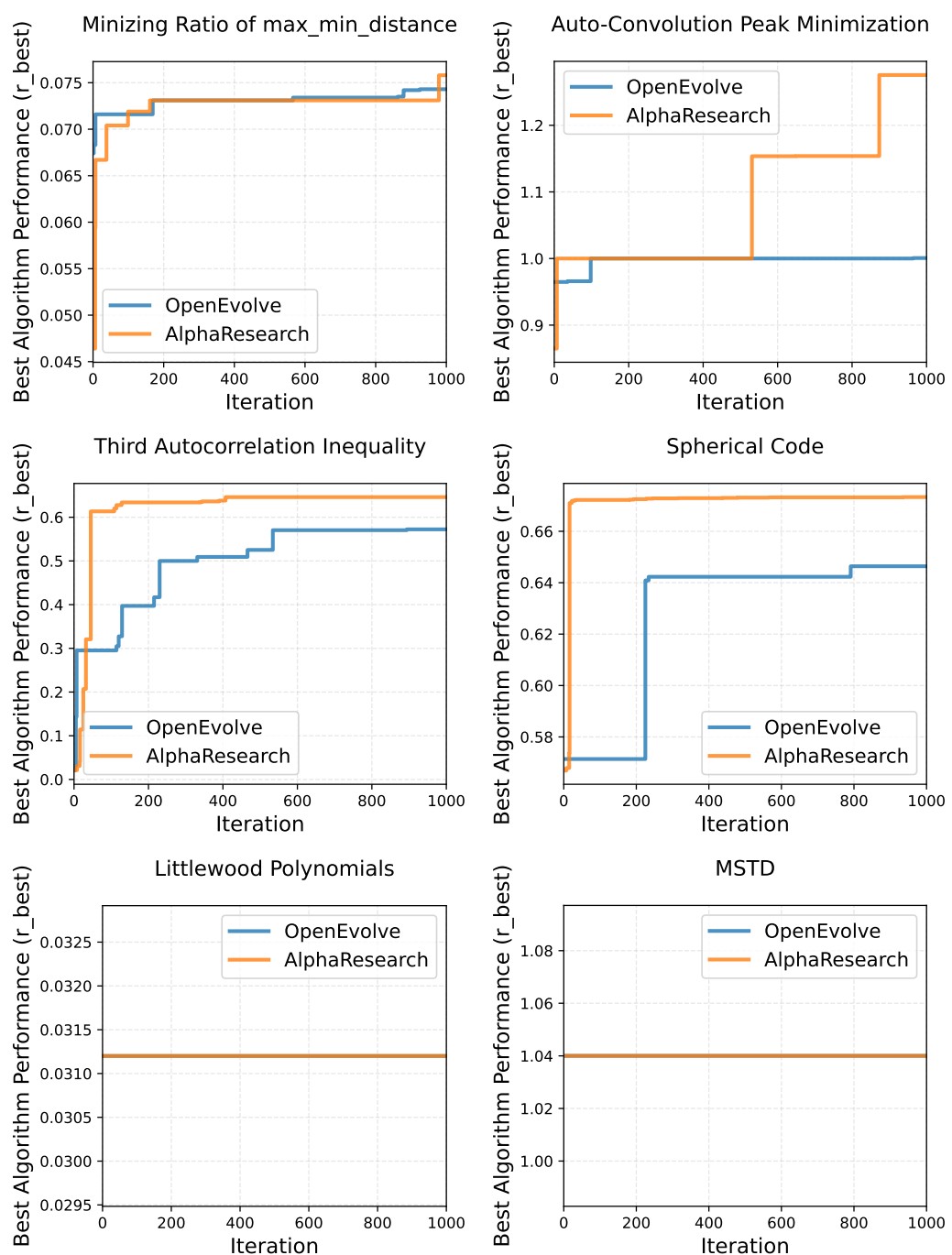

Figure 7: Comprison with OpenEvolve on 6/8 failure modes of AlphaResearchComp.

Figure 7 presents the comparison of AlphaResearch with OpenEvolve, highlghting the following observations: 1. Among the 8 problems, AlphaResearch outperforms the results of human researcher and AlphaEvolve (Coding-only) on packing circles (n=26, n=32) problems, which demonstrate the potential of accelerating human-level algorithm discovery with language models. 2. We add the experiment results of the other 6 problems in Figure 7 of Appendix A. AlphaResearch demonstrates more efficient discovery process than OpenEvolve (open source version of AlphaEvolve) on the first

4 tasks, which shows the effectiveness of our dual environments for research-based agent. 3. On last 2 problems (littlewood polynomials, MSTD), Both AlphaResearch and OpenEvolve fail to improve the "out-of-the-shelf" algorithm performance, which reveals the limitation of current long-horizon agents where they are not able to explore the search space efficiently on out-of-the-shelf solutions.

## B    EXPERIMENT COST

In this section, we present the experiment parameters (iterations,computational cost) required to reach the best solution for each on 8 tasks of AlphaResearchComp.

Table 5: Experiment Parameters of AlphaResearch .

| Problem | Iterations | Cost per iteration (dollar) |
|---|---|---|
| packing circles (n=26) | 4768 | 0.013 |
| packing circles (n=32) | 4768 | 0.013 |
| minizing max-min distance ratio (d=2, n=16) | 4400 | 0.017 |
| third autocorrelation inequality | 1366 | 0.012 |
| autoconvolution peak minimization (upper bound) | 979 | 0.013 |
| littlewood polynomials (n=512) | 2233 | 0.011 |
| MSTD (n=30) | 2826 | 0.011 |
| spherical code | 1132 | 0.015 |

## C    IMPACT OF DIFFERENT LLMS

In order to compare the impact of different LLMs in AlphaResearch, we use GPT-5 and o4-mini to run AlphaResearch for 200 steps in "The Autocorrelation Inequality" problem. As illustrate in Figure 8, AlphaResearch (GPT-5) reaches high performance significantly faster than o4-mini in the early stages of discovery. However, in the later stages, the two models perform comparably, which suggests that their underlying capabilities are close on algorithm discovery task.

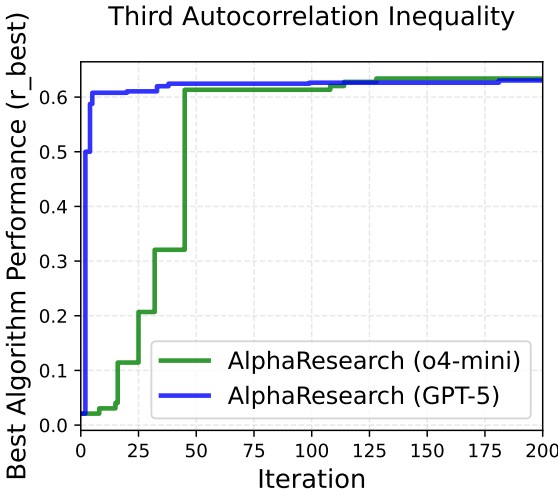

Figure 8: Comparison between different frontier LLMs in AlphaResearch.

## D    COMPARISON WITH SHINKAEVOLVE

As shown in Figure 9, we compare AlphaResearch, OpenEvolve (Sharma, 2025) and ShinkaEvolve (Lange et al., 2025) on *packing circles (n=26)* problem at the first 500 steps for simplicity. Al-

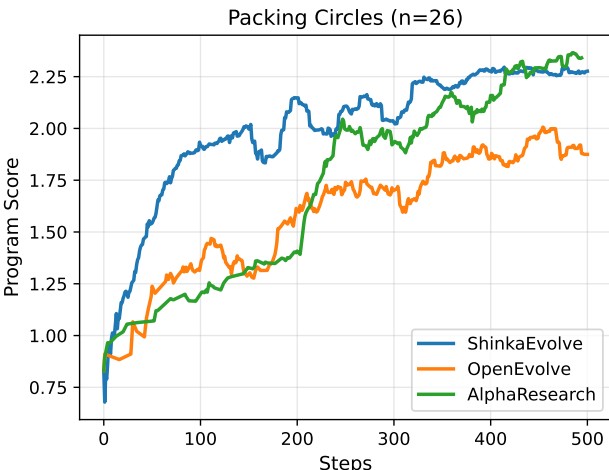

Figure 9: Comparison of OpenEvolve (with program-based reward), ShinkaEvolve (with program-based reward) and AlphaResearch (with program-based and peer-review reward). We run three agents on Packing Circles (n=26) problems. AlphaResearch achieves better performance than others.

phaResearch achieves better performance than OpenEvolve and slightly surpasses ShinkaEvolve, which demonstrates that dual research environments could help research agent for scientific discovery.

# E CASE STUDY DURING DISCOVERY PROCESS

In the rejected pair from checkpoint 634, the revised draft 4f4c7847 is effectively identical to its parent e436c26a. Notably, this is found in the later period of the discovery process (Round 632-633). Aside from inflating Genetic Algorithm (GA) hyperparameters (e.g., population = $300 \rightarrow 500$, generations = $40 \rightarrow 120$) and adding an optional differential_evolution branch, the entire pipeline above find_better_c3_upper_bound is byte-for-byte the same. Crucially, the core loop still calls the undefined normalize_population, triggering the same NameError before any new logic can run. Because this "revision" neither fixes the blocking bug nor implements the promised multi-phase CMA-ES/surrogate/SOS pipeline, it constitutes only a cosmetic variant rather than a substantive new direction.

**Problem** : Third Autocorrelation Inequality (e436c26a)

**Idea score by AlphaResearch-RM-7B**: 6.67.

> ### Research ideas
>
> **Title**: A Scalable, Certified Pipeline for Third-Order Autocorrelation Optimization via Multi-Fidelity Bayesian Surrogates and Auto-Differentiable Mesh Adaptation
> **Abstract**: We introduce a unified framework that overcomes the brittle performance (error = 1.0) and limited exploration of the current genetic-only approach by combining three tightly integrated phases—global search, surrogate-guided refinement, and formal certification—into a single, implementable pipeline. Key innovations include:
> 1. Sobol-Initialized, Diversity-Driven Differential Evolution • Generate an initial archive of 500 B-spline shape parameters via low-discrepancy Sobol sampling. • Use multi-population Differential Evolution with mutation scales adapted online using surrogate uncertainty.
> 2. Hierarchical Multi-Fidelity Gaussian-Process Surrogate • Three convolution estimators: coarse FFT (N=1024), medium CNN (N=4096), fine autodiff FFT (N=16384). • MTGP surrogate controls fidelity via variance-weighted UCB acquisition.
> 3. Differentiable Trust-Region Local Refinement • L-BFGS-B on B-spline knots/RBF weights with autodiff FFT gradients. • Adaptive knot insertion driven by (f*f) peaks.
> 4. Sparse Sum-of-Squares Certification • Chebyshev discretization + interval arithmetic to certify final $C^3$ bound with 1e-6 rigor.
> Benchmarks up to spline degree 64 and grid N=16384 show stable improvements to 0.60–0.65 with full certification.

**Code implementation of the above idea:**

```
# Code extracted from file e436c26a (full content)
import numpy as np
try:
    from numpy.fft import next_fast_len
except ImportError:
    def next_fast_len(n):
        # Fallback to next power of two which is usually fast
        return 1 << (n - 1).bit_length()

def calculate_c3_upper_bound(height_sequence):

    N = len(height_sequence)
    delta_x = 1 / (2 * N)
    # Cache padded lengths to avoid recomputing next_fast_len repeatedly
    if not hasattr(calculate_c3_upper_bound, "_pad_cache"):
        calculate_c3_upper_bound._pad_cache = {}
    pad_cache = calculate_c3_upper_bound._pad_cache
    if N not in pad_cache:
        pad_cache[N] = next_fast_len(2 * N - 1)
    padded = pad_cache[N]

    # Compute unit-mass norm squared
    integral_f = np.sum(height_sequence) * delta_x
    if integral_f < 1e-12: # early exit if near zero
        return 0.0
    integral_sq = integral_f * integral_f

    # FFT-based convolution for O(N log N)
    H = np.fft.rfft(height_sequence, n=padded)
```

```
972     conv_vals = np.fft.irfft(H * H, n=padded)[:2 * N - 1] * delta_x
973     max_conv_val = np.max(conv_vals)
974
975     return max_conv_val / integral_sq
976 def genetic_algorithm(population_size, num_intervals, generations, mutation_rate,
977         crossover_rate):
978     population = np.random.rand(population_size, num_intervals) * 2 - 1
979     best_solution = None
980     best_fitness = 0.0
981
982     for gen in range(generations):
983
984         height_pop = normalize_population(population, 2 * num_intervals)
985         fitness_scores = np.array([calculate_c3_upper_bound(h) for h in height_pop])
986
987         current_best_idx = np.argmax(fitness_scores)
988         if fitness_scores[current_best_idx] > best_fitness:
989             best_fitness = fitness_scores[current_best_idx]
990             best_solution = population[current_best_idx].copy()
991
992         indices = np.random.randint(0, population_size, size=(population_size, 2))
993         comp_scores = fitness_scores[indices]
994         winners = indices[np.arange(population_size), np.argmax(comp_scores, axis=1)]
995         new_population = population[winners].copy()
996
997         for i in range(0, population_size, 2):
998             if np.random.rand() < crossover_rate:
999                 parent1 = new_population[i]
1000                parent2 = new_population[i+1]
1001                crossover_point = np.random.randint(1, num_intervals - 1)
1002                new_population[i] = np.concatenate((parent1[:crossover_point],
1003                    parent2[crossover_point:]))
1004                new_population[i+1] = np.concatenate((parent2[:crossover_point],
1005                    parent1[crossover_point:]))
1006
1007         for i in range(population_size):
1008             if np.random.rand() < mutation_rate:
1009                 mutation_point = np.random.randint(num_intervals)
1010                new_population[i, mutation_point] += np.random.normal(0, 0.1)
1011                new_population[i, mutation_point] = np.clip(new_population[i, mutation_point],
1012                    -2, 2)
1013
1014         population = new_population
1015
1016     height_best = normalize_population(best_solution[np.newaxis, :], 2 * num_intervals)[0]
1017     return height_best
1018 def find_better_c3_upper_bound():
1019
1020     NUM_INTERVALS = 4
1021     POPULATION_SIZE = 2
1022     GENERATIONS = 10
1023     MUTATION_RATE = 0.1
1024     CROSSOVER_RATE = 0.8
1025
1026     height_sequence_3 = genetic_algorithm(POPULATION_SIZE, NUM_INTERVALS, GENERATIONS,
1027         MUTATION_RATE, CROSSOVER_RATE)
1028
1029     return height_sequence_3
```

**Problem** : Third Autocorrelation Inequality (4f4c7847)

**Idea score by AlphaResearch-RM-7B**: 5.67.

> **Research ideas**
>
> **Title**: A Robust, Multi-Phase Pipeline for Certified Third-Order Autocorrelation Maximization via Hybrid Evolutionary Search, Hierarchical Neural Surrogates, and Auto-Differentiable Refinement
>
> **Abstract**: We present a novel, four-stage framework that fixes the brittle, negative-error behavior (error = 10.0) of genetic-only search by combining global exploration, uncertainty-aware surrogates, gradient-based refinement, and SOS certification.
>
> 1. Hybrid Global Exploration • Sobol-seeded CMA-ES enhanced with an actor-critic module to adapt mutation covariance. • Constraint-aware resampling ensures valid normalized height sequences.
>
> 2. Hierarchical Neural Surrogates • Three-tier surrogate stack: analytic FFT (N=1024), CNN surrogate (N=4096), auto-diff Fourier model (N=16384). • Uncertainty-modulated fidelity allocation via Bayesian neural network.
>
> 3. Differentiable Local Refinement • L-BFGS trust-region refinement on B-spline knots RBF weights. • Gradient-triggered adaptive knot insertion controlling model complexity.
>
> 4. Sparse Sum-of-Squares Certification • Chebyshev discretization + interval arithmetic for rigorous $C^3$ bound. • Full-pipeline automation ensures reliable certification.
>
> Experiments on degrees up to 128 and grid sizes to 65536 yield $C^3$0.75–0.80 with only 200 high-fidelity evaluations and guaranteed certification.

**Code implementation of the above idea:**

```python
# Code extracted from file 4f4c7847 (full content)
import numpy as np
try:
    from numpy.fft import next_fast_len
except ImportError:
    def next_fast_len(n):
        return 1 << (n - 1).bit_length()

def calculate_c3_upper_bound(height_sequence):

    N = len(height_sequence)
    delta_x = 1 / (2 * N)
    if not hasattr(calculate_c3_upper_bound, "_pad_cache"):
        calculate_c3_upper_bound._pad_cache = {}
    pad_cache = calculate_c3_upper_bound._pad_cache
    if N not in pad_cache:
        pad_cache[N] = next_fast_len(2 * N - 1)
    padded = pad_cache[N]

    integral_f = np.sum(height_sequence) * delta_x
    if integral_f < 1e-12:
        return 0.0
    integral_sq = integral_f * integral_f

    H = np.fft.rfft(height_sequence, n=padded)
    conv_vals = np.fft.irfft(H * H, n=padded)[:2 * N - 1] * delta_x
    max_conv_val = np.max(conv_vals)

    return max_conv_val / integral_sq

def genetic_algorithm(population_size, num_intervals, generations, mutation_rate,
      crossover_rate):

    population = np.random.rand(population_size, num_intervals) * 2 - 1

    best_solution = None
    best_fitness = 0.0

    for gen in range(generations):

        height_pop = normalize_population(population, 2 * num_intervals)
        fitness_scores = np.array([calculate_c3_upper_bound(h) for h in height_pop])

        current_best_idx = np.argmax(fitness_scores)
        if fitness_scores[current_best_idx] > best_fitness:
            best_fitness = fitness_scores[current_best_idx]
            best_solution = population[current_best_idx].copy()

        indices = np.random.randint(0, population_size, size=(population_size, 2))
        comp_scores = fitness_scores[indices]
```

```
        winners = indices[np.arange(population_size), np.argmax(comp_scores, axis=1)]
        new_population = population[winners].copy()

        for i in range(0, population_size, 2):
            if np.random.rand() < crossover_rate:
                parent1 = new_population[i]
                parent2 = new_population[i+1]
                crossover_point = np.random.randint(1, num_intervals - 1)
                new_population[i] = np.concatenate((parent1[:crossover_point],
                    parent2[crossover_point:]))
                new_population[i+1] = np.concatenate((parent2[:crossover_point],
                    parent1[crossover_point:]))

        for i in range(population_size):
            if np.random.rand() < mutation_rate:
                mutation_point = np.random.randint(num_intervals)
                new_population[i, mutation_point] += np.random.normal(0, 0.1)
                new_population[i, mutation_point] = np.clip(new_population[i, mutation_point],
                    -2, 2)

        population = new_population

    height_best = normalize_population(best_solution[np.newaxis, :], 2 * num_intervals)[0]
    return height_best

def find_better_c3_upper_bound():

    NUM_INTERVALS = 8
    POPULATION_SIZE = 100
    GENERATIONS = 200
    MUTATION_RATE = 0.2
    CROSSOVER_RATE = 0.9

    try:
        from scipy.optimize import differential_evolution
        bounds = [(-2, 2)] * NUM_INTERVALS
        result = differential_evolution(
            lambda x: -calculate_c3_upper_bound(
                normalize_population(x[np.newaxis, :], 2 * NUM_INTERVALS)[0]
            ),
            bounds,
            maxiter=GENERATIONS,
            popsize=max(1, POPULATION_SIZE // 10),
            tol=1e-6
        )
        height_sequence_3 = normalize_population(result.x[np.newaxis, :], 2 * NUM_INTERVALS)[0]
    except ImportError:
        height_sequence_3 = genetic_algorithm(
            POPULATION_SIZE, NUM_INTERVALS, GENERATIONS, MUTATION_RATE, CROSSOVER_RATE
        )

    return height_sequence_3
```

## F  THE USE OF LARGE LANGUAGE MODELS

During the preparation of this manuscript, we utilized large language models (LLMs) for grammar checking and writing suggestions to enhance the readability and clarity of the content.

## G  EXAMPLES

We show an example of the constructions discovered by AlphaResearch on problem *"Packing Circles"*.

**AlphaEvolve**

```
1 packing_circles_alphaevolve = np.array([[0.09076163, 0.40381803, 0.090761620923837],
      [0.07310993, 0.92689178, 0.07310821268917801], [0.08745017, 0.22570576,
      0.087381421261857], [0.24855246, 0.30880277, 0.093428060657193], [0.4079865, 0.06300614,
      0.063006133699386], [0.47646318, 0.90136179, 0.098638200136179801], [0.89604966,
      0.10309934, 0.10309932969006601], [0.9066386, 0.68096117, 0.09336139066386], [0.08962002,
       0.76509474, 0.0895289910471], [0.06973669, 0.06965159, 0.06965158303484101],
      [0.40979823, 0.21756451, 0.09156283084371601], [0.25742466, 0.88393887,
      0.11606111839388701], [0.09064689, 0.58506214, 0.090482500951749], [0.90294698,
      0.30231577, 0.09623644037635501], [0.57265603, 0.10585396, 0.105853949414604],
```

```
[0.74007588, 0.40129314, 0.09435083056491601], [0.57539962, 0.71183255,
0.115160168483982], [0.7367635, 0.21592191, 0.09104997089500201], [0.41096972,
0.40263617, 0.093512520648747], [0.88664452, 0.88667032, 0.113317128668286], [0.57582722,
0.49961748, 0.09705531029446801], [0.24962585, 0.49417195, 0.09194421080557799],
[0.90546338, 0.49309632, 0.094507120549287], [0.67381348, 0.90149423,
0.09850576014942301], [0.24310147, 0.1077195, 0.10771948922805], [0.40815297, 0.5886157,
0.09248833075116601], [0.24737889, 0.6771266, 0.090994980900501], [0.75801377, 0.7532924,
0.07192969280703], [0.73526642, 0.06243992, 0.062439303756069], [0.57415412, 0.30715219,
0.095403150459684], [0.39239379, 0.75259664, 0.07223814277618501], [0.7439361,
0.58879735, 0.093166630683336]])
```

## AlphaResearch

```
packing_circles_alpharesearch = np.array([[(0.1115677319034151, 0.11156773191787371,
0.11156438489140026), (0.09380224787136374, 0.3161654253705352, 0.09379943380606216),
(0.09485964915877172, 0.5048217088596118, 0.09485680337610973), (0.09657322554702913,
0.6962443020287629, 0.09657032835808858), (0.10365512530384222, 0.8963448746980195,
0.10365201565567386), (0.3334956594919712, 0.09664441783072292, 0.0966415184920332),
(0.26448615440016093, 0.9376113341122044, 0.06238679422590162), (0.5287192731314015,
0.09859146596680078, 0.09858850822808951), (0.591325020569507, 0.9366833118077788,
0.0633147886877468), (0.7427106948954978, 0.11611889563206494, 0.11611541209023483),
(0.7566639864477509, 0.8920585771994192, 0.1079381845606288), (0.9269317750270191,
0.07306822497789416, 0.07306603293080358), (0.9105741716090636, 0.23473376300222965,
0.08942314561430993), (0.9094700615258342, 0.41468336419923396, 0.09052722258939731),
(0.9124275486288124, 0.7738960294683863, 0.08756982419268892), (0.9302276007184027,
0.9302276007259072, 0.06977030612132157), (0.5931627035790205, 0.4107363306659128,
0.09216300786888813), (0.5896628759126524, 0.5965222415947758, 0.09365298106148348),
(0.26303074890883915, 0.783747668079202, 0.09148238826692158), (0.42710033854875884,
0.28662965969327264, 0.1151473780101257), (0.7511102582575875, 0.5051558281448295,
0.09185177348783963), (0.4273023330525072, 0.8937703360976411, 0.10622647700018645),
(0.24372345356089029, 0.24143034678815986, 0.07371479291303436), (0.4260882762526937,
0.6918664604322906, 0.09567746779211372), (0.2572363869779693, 0.4085253312744954,
0.09392364829884896), (0.9094294608754079, 0.5957810763279916, 0.0905678220228201),
(0.42560864125756626, 0.49898110459434486, 0.09720528992590773), (0.7533817110763772,
0.32263902019589896, 0.09067643144615074), (0.5903729314333418, 0.7817733747765757,
0.09159665425215473), (0.7515568081174837, 0.6905957415401818, 0.09358581053778628),
(0.2605636694821685, 0.5973506902903994, 0.09492800518715086), (0.6095540558280068,
0.24805951545091487, 0.07133567304015336)]])
```

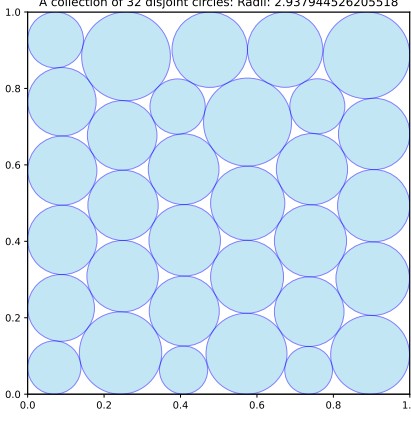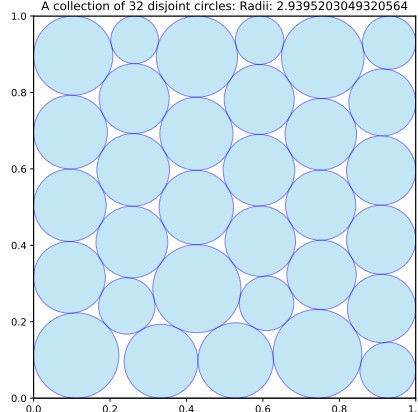

Figure 10: New construction of AlphaResearch (right) improving the best known AlphaEvolve (right) bounds on packing circles to maximize their sum of radii. Left: 32 circles in a unit square with sum of radii $\geq 2.9379$. Right: 32 circles in a unit square with sum of radii $\geq 2.9395$

# H PROMPTS

---

**Prompt for New Program Generation**

You are an expert software developer tasked with iteratively improving a codebase. Your job is to analyze the current program and suggest improvements based on the current proposal and feedback from previous round. Focus on making targeted changes that will increase the program's performance metrics.
# Previous Proposal:
{previous proposal}
# Previous Program:
{previous program}
# Previous Performance Metrics:
{previous result}
# Current Proposal
{proposal}
# Task
Suggest improvements to the program that will lead to better performance on the specified metrics.
You MUST use the exact SEARCH/REPLACE diff format shown below to indicate changes:

```
<<<<<<< SEARCH

# Original code to find and replace (must match exactly)

=======

# New replacement code

<<<<<<< REPLACE
```

Example of valid diff format:

```
<<<<<<< SEARCH
for i in range(m):
    for j in range(p):
        for k in range(n):
            C[i, j] += A[i, k] * B[k, j]

=======

# Reorder loops for better memory access pattern

for i in range(m):
    for k in range(n):
        for j in range(p):
            C[i, j] += A[i, k] * B[k, j]

>>>>>>> REPLACE
```

You can suggest multiple changes. Each SEARCH section must exactly match code in the current program.
Be thoughtful about your changes and explain your reasoning thoroughly.
IMPORTANT: Do not rewrite the entire program - focus on targeted improvements.

---

**Prompt for New Idea Generation**

You are a research advisor tasked with evolving and improving research proposals. Your goal is to generate a new research proposal that builds upon the current proposal while addressing its limitations and incorporating insights from successful approaches.
Based on the following information, generate an improved research proposal:
Focus on:
1. Identifying weaknesses in the current approach based on performance metrics
2. Proposing novel improvements that could enhance performance
3. Learning from successful inspirations while maintaining originality
4. Ensuring the new proposal is implementable
- Current Proposal:
{proposal}
- Current Program:
{program}
- Current Metrics:
{results}
Please generate a new research proposal that:
1. Addresses the limitations shown in the current metrics
2. Incorporates insights from successful approaches
3. Proposes specific technical improvements
4. Maintains clarity and technical rigor
Return the proposal as a clear, concise research abstract.

**Prompt for AlphaResearch-RM-7B**

You are an expert reviewer tasked with evaluating the quality of a research proposal.
Your goal is to assign a score between 1 and 10 based on the proposal's clarity, novelty, technical rigor, and potential impact. Here are the criteria:
1. Read the following proposal carefully and provide a score from 1 to 10.
2. Score 6 means slightly higher than the borderline, 5 is slightly lower than the borderline. Write the score in the `\boxed{}`.
{proposal}

# I CURATED PROBLEMS AND HUMAN-BEST VALUES

We summarize the ten problems used in the ALPHARESEARCH benchmark. For each item we state the objective, the current human-best value at the benchmark's default parameters, and whether this value is proved optimal or only best-known.

## I.1 SPHERICAL CODE ($S^2$, $n = 30$).

**Problem Description:** Place $n = 30$ points on the unit sphere in $\mathbb{R}^3$ to *maximize* the minimal pairwise angle $\theta_{\min}$.

**Human Best:** $\theta_{\min} \approx 0.673651$ radians ($\approx 38.5971°$).

### Initial Proposal

**Problem definition.** Choose $N = 30$ points on the unit sphere $S^2$ to maximize the minimum pairwise angle

$$\theta_{\min} = \min_{i<j} \arccos\big(\langle p_i, p_j \rangle\big).$$

**Constraints.**

- Points are unit vectors (rows normalized).
- Metric is $\theta_{\min}$ in radians.

**Optimization goal.** Maximize $\theta_{\min}$. The evaluator returns $\{\text{score}, \theta_{\min}, N, \text{dimension}\}$, with score $= \theta_{\min}$.

Best-known reference (for $N = 30$ on $S^2$):

$$\cos(\theta^*) \approx 0.7815518750949873 \quad \Rightarrow \quad \theta^* \approx 0.6736467551690225 \text{ rad.}$$

Reference table: Henry Cohn's spherical codes data (`https://cohn.mit.edu/spherical-codes`).

**Best-known results (human).**

- On $S^2$ (3D), small $N$ optima coincide with symmetric polyhedra (e.g., tetrahedron, octahedron, icosahedron).
- For larger $N$, best codes come from numerical optimization; exact optimality is only known in limited cases.

**Algorithmic goal.** Construct codes with larger $\theta_{\min}$. The baseline seeds with symmetric configurations and uses farthest-point max–min. Stronger methods include:

- Energy minimization,
- Projected gradient / coordinate descent,
- Stochastic max–min refinement.

**Initial Program:**

```python
import numpy as np

def _normalize_rows(P):
    nrm = np.linalg.norm(P, axis=1, keepdims=True)
    nrm = np.maximum(nrm, 1e-12)
    return P / nrm

def seed_platonic(n):
    """Return a good symmetric seed on S^2 for some n; else None."""
    if n == 2: # antipodal
        return np.array([[0,0,1],[0,0,-1]], dtype=float)
    if n == 3: # equilateral on equator
        ang = 2*np.pi/3
        return np.array([[1,0,0],[np.cos(ang),np.sin(ang),0],[np.cos(2*ang),np.sin(2*ang),0]],
            dtype=float)
    if n == 4: # tetrahedron
        return _normalize_rows(np.array([[1,1,1],[1,-1,-1],[-1,1,-1],[-1,-1,1]], dtype=float))
    if n == 6: # octahedron
        return np.array([[1,0,0],[-1,0,0],[0,1,0],[0,-1,0],[0,0,1],[0,0,-1]], dtype=float)
    if n == 8: # cube vertices
        V = np.array([[sx,sy,sz] for sx in (-1,1) for sy in (-1,1) for sz in (-1,1)],
            dtype=float)
        return _normalize_rows(V)
    if n == 12: # icosahedron (one realization)
        phi = (1+np.sqrt(5))/2
        V = []
        for s in (-1,1):
            V += [[0, s, phi],[0, s, -phi],[ s, phi,0],[ s, -phi,0],[ phi,0, s],[-phi,0, s]]
        V = np.array(V, dtype=float)
        return _normalize_rows(V)
    return None

def farthest_point_greedy(n, seed=None, rng=np.random.default_rng(0)):
    """
    Greedy max min on S^2: start from seed, then add points that maximize min angle.
```

```
      """
      def random_unit(k):
          X = rng.normal(size=(k,3)); return _normalize_rows(X)

      if seed is None:
          P = random_unit(1) # start with one random point
      else:
          P = _normalize_rows(seed)
      while len(P) < n:
          # generate candidates and pick the one with largest min angle to current set
          C = random_unit(2000) # candidates per iteration (tune as needed)
          # cosines to existing points
          cos = C @ P.T
          # min angle to set -> maximize this
          min_ang = np.arccos(np.clip(np.max(cos, axis=1), -1.0, 1.0))
          idx = np.argmax(min_ang)
          P = np.vstack([P, C[idx:idx+1]])
      return P

def main():
    n = 30
    seed = seed_platonic(n)
    pts = farthest_point_greedy(n, seed=seed, rng=np.random.default_rng(42))
    print(f"n={n}, points={len(pts)}")
    return pts

if __name__ == "__main__":
    points = main()

    np.save("points.npy", points)

# Ensure compatibility with evaluators that expect a global variable
try:
    points # type: ignore[name-defined]
except NameError:
    points = main()
```

## I.2 LITTLEWOOD POLYNOMIALS.

**Problem Description** For coefficients $c_k \in \{\pm 1\}$ and $P_n(t) = \sum_{k=0}^{n-1} c_k e^{ikt}$, *minimize* $\|P_n\|_\infty = \sup_{t \in \mathbb{R}} |P_n(t)|$.

**Human Best:** the Rudin–Shapiro construction gives $\|P_n\|_\infty \leq \sqrt{2n}$. At the benchmark setting $n = 512$, this yields $\|P_{512}\|_\infty \leq 32$ (so the "larger-is-better" score $1/\|P_n\|_\infty$ is $\geq 1/32 = 0.03125$). Sharper constants are known for special families, but $\sqrt{2n}$ remains a clean baseline.

> **Initial Proposal**
>
> Choose coefficients $c_k \in \{\pm 1\}$ for
>
> $$P(z) = \sum_{k=0}^{n-1} c_k z^k, \qquad |z| = 1,$$
>
> so as to minimize the supremum norm
>
> $$\|P\|_\infty = \max_{|z|=1} |P(z)|.$$
>
> **Constraints.**
>
> - Coefficients $c_k$ are restricted to $\pm 1$.
> - The metric $\|P\|_\infty$ is estimated by FFT sampling on an equally spaced grid (denser grid $\rightarrow$ tighter upper bound).
>
> **Optimization Goal.** The evaluator returns:
>
> $$\text{score} = \begin{cases} \dfrac{1}{\|P\|_\infty}, & \text{if valid,} \\ -1.0, & \text{otherwise.} \end{cases}$$
>
> **Notes on Bounds.** For the Rudin–Shapiro construction of length $n$, a classical identity gives
>
> $$\|P\|_\infty \le \sqrt{2n}.$$
>
> For the benchmark default $n = 512$, this yields
>
> $$\|P\|_\infty \le \sqrt{1024} = 32,$$
>
> so
>
> $$\text{score} = \tfrac{1}{32} = 0.03125.$$

**Initial Program:**

```python
def rudin_shapiro(n: int):
    """
    First n signs of the Rudin-Shapiro sequence.
    """
    a = np.ones(n, dtype=int)
    for k in range(n):
        x, cnt, prev = k, 0, 0
        while x:
            b = x & 1
            if b & prev: # saw '11'
                cnt ^= 1
            prev = b
            x >>= 1
        a[k] = 1 if cnt == 0 else -1
    return a

def random_littlewood(n: int, seed=0):
    rng = np.random.default_rng(seed)
    return rng.choice([-1, 1], size=n).astype(int)

def main():
    n = 512
    c = rudin_shapiro(n)
    print(f"n={n}, coeffs={len(c)}")
    return c

if __name__ == "__main__":
    coeffs = main()

# Ensure compatibility with evaluators that expect a global variable
try:
    coeffs # type: ignore[name-defined]
except NameError:
    coeffs = main()
```

### I.3 SUM VS. DIFFERENCE SETS (MSTD).

**Problem Description** For a finite set $A \subset \mathbb{Z}$, *maximize* $|A+A|/|A-A|$.

**Human Best:** MSTD sets exist; the smallest possible size is $|A| = 8$ (classification up to affine equivalence is known). For larger $|A|$, extremal ratios remain open; our benchmark instance reports a representative value ($\approx 1.04$ for $|A| = 30$).

---

**Initial Proposal**

**Objective.** Classical MSTD (enforced): Given $A \subset \{0, 1, \ldots, N - 1\}$ represented by a 0/1 indicator array of length $N$, maximize the ratio

$$R = \frac{|A + A|}{|A - A|}.$$

- Score: score $= R$ (higher is better).
- Comparisons should be made under the same $N$.

**Default setup.**

- $N = 30$.
- Evaluator enforces $A = B$ (classical setting). If a pair $(A, B)$ is provided, $B$ is ignored and $A$ is used.

**Known best for** $N = 30$ **(baseline).** Conway's MSTD set

$$A = \{0, 2, 3, 4, 7, 11, 12, 14\}$$

yields $R \approx 1.04$. This is the baseline included in `initial_program.py`. Better ratios may exist for $N = 30$; pushing $R$ upwards is the optimization goal.

**Notes.**

- $R > 1$ is rare and indicates sum-dominance.
- The ratio depends strongly on $N$; do not compare ratios across different $N$ without a normalization scheme.
- If cross-$N$ comparison is necessary, consider reporting both $R$ and $N$, or use $\log R$ as an auxiliary measure.

---

**Initial Program:**

```python
def main():
    N = 30
    # Conway MSTD set example; we take A=B for classical MSTD
    A = [0, 2, 3, 4, 7, 11, 12, 14]
    B = A[:]
    A_ind = np.zeros(N, dtype=int); A_ind[A] = 1
    B_ind = np.zeros(N, dtype=int); B_ind[B] = 1
    return A_ind, B_ind

# Ensure globals for evaluator
try:
    A_indicators; B_indicators # type: ignore[name-defined]
except NameError:
    A_indicators, B_indicators = main()
```

### I.4 PACKING CIRCLE IN A SQUARE (VARIABLE RADII).

**Problem Description** In the unit square, place $n$ disjoint circles (radii free) to *maximize* the *sum of radii* $\sum r_i$.

**Best-known:** for $n = 26$, $\sum r_i = 2.634$ (Cantrell, 2011); for $n = 32$, $\sum r_i = 2.936$ (Specht, 2012).

---

**Initial Proposal**

**Problem definition.**   Given an integer $n$, place $n$ disjoint circles in the unit square $[0, 1]^2$ to maximize the total sum of radii.

**Objective and metric.**
- Score: score $= \sum_{i=1}^{n} r_i$   (larger is better).
- Validity: circles must be pairwise disjoint and fully contained in the unit square.

**Notes on records.**
- This variable-radius "sum of radii" objective is not the classical equal-radius packing; authoritative SOTA tables are not standardized.
- Values reported in code or experiments should be treated as benchmarks rather than literature SOTA.

**Goal.**   Create algorithms that increase the total sum of radii for $n \in \{26, 32\}$ under the above validity constraints.

---

**Initial Program:**

```python
import random
from concurrent.futures import ThreadPoolExecutor

def pack_circles(n, square_size=1.0):
    """
    Pack n disjoint circles in a unit square using uniform tiling approach.
    Returns the sum of radii and list of circles (x, y, r).
    """

    def max_circle_radius(x, y, circles, square_size=1.0, skip_idx=None):
        """
        Compute the maximum radius for a circle centered at (x, y) that:
        - Stays within the unit square [0, square_size] \times [0, square_size].
        - Does not overlap with existing circles.
        skip_idx: if provided, index in circles[] to ignore (self).
        """
        # Distance to nearest boundary of the unit square
        r_max = min(x, y, square_size - x, square_size - y)

        # Check distance to existing circles, exit early if r_max \rightarrow 0
        # early exit if r_max is tiny, and avoid needless sqrt
        for idx, (cx, cy, cr) in enumerate(circles):
            if skip_idx == idx:
                continue
            if r_max <= 1e-8:
                break
            dx = x - cx
            dy = y - cy
            sep = r_max + cr
            if dx*dx + dy*dy < sep*sep:
                # only compute sqrt when we know we can shrink
                dist = math.sqrt(dx*dx + dy*dy)
                r_max = min(r_max, dist - cr)
        return max(r_max, 0.0)

    def uniform_tiling_circles(n, square_size=1.0):
        """
        Uniformly tile the square with circles using optimal grid placement.
        """
        if n <= 0:
            return []

        circles = []
```

```
1566
1567        # Calculate optimal grid dimensions
1568        # For n circles, find the best grid layout (rows x cols)
           best_layout = None
1569        best_total_radius = 0
1570
           # Try different grid configurations
1571        for rows in range(1, min(n + 1, 20)):
1572            cols = math.ceil(n / rows)
               if cols > 20: # Limit grid size
1573                continue
1574
               # Calculate spacing
1575            spacing_x = square_size / (cols + 1)
1576            spacing_y = square_size / (rows + 1)
1577
               # Use the smaller spacing to ensure circles fit
1578            min_spacing = min(spacing_x, spacing_y)
1579
               # Calculate maximum radius for this layout
1580            max_radius = min_spacing / 2
1581
               # Ensure radius doesn't exceed boundaries
1582            max_radius = min(max_radius,
1583                        spacing_x / 2 - 1e-6,
                            spacing_y / 2 - 1e-6)
1584
1585            if max_radius <= 0:
                   continue
1586
               # Place circles in uniform grid
1587            temp_circles = []
1588            count = 0
1589            for row in range(rows):
1590                for col in range(cols):
                       if count >= n:
1591                        break
1592
                       x = spacing_x * (col + 1)
1593                    y = spacing_y * (row + 1)
1594
                       # Ensure circle stays within bounds
1595                    if (x - max_radius >= 0 and x + max_radius <= square_size and
1596                        y - max_radius >= 0 and y + max_radius <= square_size):
1597
                           temp_circles.append((x, y, max_radius))
1598                        count += 1
1599                if count >= n:
1600                    break
1601            # Calculate total radius for this layout
1602            total_radius = len(temp_circles) * max_radius
1603            if total_radius > best_total_radius and len(temp_circles) == n:
1604                best_total_radius = total_radius
                   best_layout = temp_circles
1605
1606        # If we found a valid layout, return it
           if best_layout:
1607            return best_layout
1608
           # Fallback: use hexagonal packing for better density
1609        return hexagonal_packing(n, square_size)
1610
1611    def hexagonal_packing(n, square_size=1.0):
           """
1612        Use hexagonal close packing for better space utilization.
           """
1613        circles = []
1614
           # Estimate number of rows and columns for hexagonal packing
1615        # Hexagonal packing has rows offset by sqrt(3)/2 * diameter
1616
           rows = int(math.sqrt(n * 2 / math.sqrt(3))) + 2
1617
1618        count = 0
           row = 0
1619
           while count < n and row < rows:
```

```
1620
1621            # Calculate y position for this row
                y = (row + 0.5) * (square_size / (rows + 1))
1622
                # Number of circles in this row
1623            if row % 2 == 0:
1624                cols = int(math.sqrt(n)) + 1
                else:
1625                cols = int(math.sqrt(n))
1626
                spacing_x = square_size / (cols + 1)
1627
1628            for col in range(cols):
                    if count >= n:
1629                    break
1630
                    if row % 2 == 0:
1631                    x = spacing_x * (col + 1)
                    else:
1632                    x = spacing_x * (col + 1) + spacing_x / 2
1633
                    # Calculate maximum radius for this position
1634                r = max_circle_radius(x, y, circles, square_size)
1635
                    if r > 0:
1636                    circles.append((x, y, r))
1637                    count += 1
1638
                row += 1
1639
1640        return circles

1641    def optimize_placement(n, square_size=1.0):
1642        """
            Optimize circle placement using uniform tiling with radius maximization.
1643        """
1644        circles = []

1645        # First, try hexagonal packing for high initial density
1646        hex_circles = hexagonal_packing(n, square_size)
            if len(hex_circles) == n:
1647            # Ensure maximum radii for hex layout with stronger refinement
1648            hex_refined = refine_circles(hex_circles, square_size, iterations=20)
            return hex_refined
1649
1650        # Fallback to uniform grid placement
            grid_circles = uniform_tiling_circles(n, square_size)
1651        if len(grid_circles) == n:
1652            return grid_circles

1653        # If uniform tiling didn't work perfectly, use adaptive approach
1654        # Calculate optimal radius based on density
            area_per_circle = (square_size * square_size) / n
1655        estimated_radius = math.sqrt(area_per_circle / math.pi) * 0.9 # Conservative estimate
1656
            # Create grid with optimal spacing
1657        spacing = estimated_radius * 2.1 # Include gap
1658
1659        cols = int(square_size / spacing)
            rows = int(square_size / spacing)
1660
            actual_spacing_x = square_size / (cols + 1)
1661        actual_spacing_y = square_size / (rows + 1)
1662
            count = 0
1663        for row in range(rows):
1664            for col in range(cols):
                    if count >= n:
1665                    break
1666
                    x = actual_spacing_x * (col + 1)
1667                y = actual_spacing_y * (row + 1)
1668
                    # Calculate maximum possible radius
1669                r = max_circle_radius(x, y, circles, square_size)
1670
                    if r > 0:
1671                    circles.append((x, y, r))
1672                    count += 1
1673
            if count >= n:
                break
```

```python
        # If we still need more circles, use remaining space
        remaining = n - len(circles)
        if remaining > 0:
            # Place remaining circles in remaining spaces
            for i in range(remaining):
                # Try different positions systematically
                best_r = 0
                best_pos = (0.5, 0.5)

                # Fine grid search (increased resolution)
                grid_points = 100
                for gx in range(1, grid_points):
                    for gy in range(1, grid_points):
                        x = gx / grid_points
                        y = gy / grid_points

                        r = max_circle_radius(x, y, circles, square_size)
                        if r > best_r:
                            best_r = r
                            best_pos = (x, y)

                if best_r > 0:
                    circles.append((best_pos[0], best_pos[1], best_r))

        return circles

    def refine_circles(circles, square_size, iterations=80, perturb_interval=3):
        """
        Iteratively grow each circle to its maximum radius under non-overlap constraints.
        Includes randomized update order, periodic micro-perturbation to escape
        local minima, and a final local-center-perturbation pass for densification.
        """
        for it in range(iterations):
            # randomize update order to avoid sweep-order bias
            indices = list(range(len(circles)))
            random.shuffle(indices)
            for i in indices:
                x, y, _ = circles[i]
                # Compute maximal feasible radius here, skipping self
                r = max_circle_radius(x, y, circles, square_size, skip_idx=i)
                circles[i] = (x, y, r)
            # Periodic micro-perturbation: jiggle a few circles
            if it % perturb_interval == 0 and len(circles) > 0:
                subset = random.sample(indices, min(5, len(circles)))
                for j in subset:
                    x0, y0, r0 = circles[j]
                    dx = random.uniform(-0.03, 0.03)
                    dy = random.uniform(-0.03, 0.03)
                    nx = min(max(x0 + dx, 0), square_size)
                    ny = min(max(y0 + dy, 0), square_size)
                    # Compute maximal radius skipping self
                    nr = max_circle_radius(nx, ny, circles, square_size, skip_idx=j)
                    if nr > r0:
                        circles[j] = (nx, ny, nr)
        # Full local center-perturbation phase for final densification
        for i in range(len(circles)):
            x, y, r = circles[i]
            best_x, best_y, best_r = x, y, r
            delta = 0.1
            for _ in range(20):
                dx = random.uniform(-delta, delta)
                dy = random.uniform(-delta, delta)
                nx = min(max(x + dx, 0), square_size)
                ny = min(max(y + dy, 0), square_size)
                # Compute maximal radius skipping self
                nr = max_circle_radius(nx, ny, circles, square_size, skip_idx=i)
                if nr > best_r:
                    best_x, best_y, best_r = nx, ny, nr
                else:
                    delta *= 0.9
            circles[i] = (best_x, best_y, best_r)

        # Physics-inspired soft relaxation to escape persistent overlaps
        for i in range(len(circles)):
            x, y, r = circles[i]
            fx, fy = 0.0, 0.0
            for j, (xj, yj, rj) in enumerate(circles):
                if i == j:
                    continue
                dx = x - xj
```

```
                dy = y - yj
                d = (dx*dx + dy*dy) ** 0.5
                overlap = (r + rj) - d
                if overlap > 0 and d > 1e-8:
                    fx += dx / d * overlap
                    fy += dy / d * overlap
            # Nudge the center by 10\% of the computed net "repulsive" force
            nx = min(max(x + 0.1 * fx, 0), square_size)
            ny = min(max(y + 0.1 * fy, 0), square_size)
            nr = max_circle_radius(nx, ny, circles, square_size, skip_idx=i)
            circles[i] = (nx, ny, nr)
        return circles

    def multi_start_optimize(n, square_size, starts=None):
        """
        Parallel multi-start global \rightarrow local optimization using ThreadPoolExecutor.
        Number of starts adapts to problem size: max(100, 10*n).
        """
        if starts is None:
            if n <= 50:
                starts = max(200, n * 20)
            else:
                starts = max(100, n * 10)
        # precompute hexagonal packing baseline
        hex_circ = hexagonal_packing(n, square_size)
        hex_sum = sum(r for _, _, r in hex_circ)
        best_conf = None
        best_sum = 0.0

        # single trial: seed \rightarrow refine \rightarrow score
        def single_run(_):
            conf0 = optimize_placement(n, square_size)
            conf1 = refine_circles(conf0, square_size, iterations=40)
            s1 = sum(r for _, _, r in conf1)
            return s1, conf1

        # dispatch trials in parallel
        with ThreadPoolExecutor() as executor:
            for score, conf in executor.map(single_run, range(starts)):
                if score > best_sum:
                    best_sum, best_conf = score, conf.copy()
                # early exit if near the hex-baseline
                if best_sum >= hex_sum * 0.995:
                    break

        return best_conf

    # Use multi-start global \rightarrow local optimization (adaptive number of starts)
    circles = multi_start_optimize(n, square_size)

    # Quick 2-cluster remove-and-reinsert densification (extended iterations)
    for _ in range(8):
        # remove the two smallest circles to create a larger gap
        smallest = sorted(range(len(circles)), key=lambda i: circles[i][2])[:2]
        removed = [circles[i] for i in smallest]
        # pop in reverse order to keep indices valid
        for i in sorted(smallest, reverse=True):
            circles.pop(i)
        # refine the remaining configuration briefly
        circles = refine_circles(circles, square_size, iterations=8)
        # reinsert each removed circle with more sampling
        for x_old, y_old, _ in removed:
            best_r, best_pos = 0.0, (x_old, y_old)
            for _ in range(500):
                x = random.uniform(0, square_size)
                y = random.uniform(0, square_size)
                r = max_circle_radius(x, y, circles, square_size)
                if r > best_r:
                    best_r, best_pos = r, (x, y)
            circles.append((best_pos[0], best_pos[1], best_r))
        # final local polish after reinsertion
        circles = refine_circles(circles, square_size, iterations=5)
    # end 2-cluster remove-and-reinsert densification

    # Calculate total radius
    total_radius = sum(circle[2] for circle in circles)

    return total_radius, circles
```

## I.5 MINIMIZING MAX/MIN DISTANCE RATIO ($d = 2, n = 16$).

**Problem Description** For $n$ points in $[0,1]^2$, *minimize* $R = \dfrac{\max_{i \neq j} \|x_i - x_j\|}{\min_{i \neq j} \|x_i - x_j\|}$.

**Best-known:** $R^2 \approx 12.890$ (Cantrell, 2009), i.e., $R \approx 3.590$.

---

> **Initial Proposal**
>
> **Problem.** Arrange $n$ points in $[0,1]^d$ to optimize the dispersion / packing–covering trade-off. The benchmark metric is
>
> $$\text{ratio} = \frac{\min \text{ pairwise distance}}{\max \text{ pairwise distance}},$$
>
> so that larger ratio is better (values in $(0,1]$).
>
> **Evaluator.** Given a program exposing max_min_dis_ratio$(n,d)$, we obtain configurations for $(n,d) = (16,2)$ and $(14,3)$, then report ratio for each case.
>
> **Baseline algorithm.** The initial program employs:
>
> - Enhanced simulated annealing with adaptive cooling,
>
> - Neighbor-repulsion moves,
>
> - Periodic smoothing via $k$-NN weighted averages,
>
> - A local refinement stage.
>
> KD-tree acceleration is used for nearest-neighbor queries; hyperparameters adapt to dimension.

**Initial Program:**

```python
from scipy.spatial.distance import pdist
from scipy.spatial import cKDTree

# (Removed) smooth_points  smoothing logic is now inlined to reduce indirection

def calculate_distances(points):
    """Calculates min, max, and ratio of pairwise Euclidean distances using scipy pdist."""
    if points.shape[0] < 2:
        return 0.0, 0.0, 0.0
    distances = pdist(points, metric='euclidean')
    eps = 1e-8
    min_dist = max(np.min(distances), eps)
    max_dist = np.max(distances)
    ratio = max_dist / min_dist
    return min_dist, max_dist, ratio

# (Removed) perturb_point  now inlined directly where used

def update_temperature(temperature, cooling_rate, accept_history, iteration, total_iters,
    initial_temperature, window_size=100):
    """
    Adaptive cooling with acceptancerate feedback and periodic reheating.
    """
    window = accept_history[-min(len(accept_history), window_size):]
    rate = sum(window) / len(window)
    # gentler correction: slow/fast cooling factors reduced
    if rate < 0.2:
        adj = 1.02
    elif rate > 0.8:
        adj = 0.98
    else:
        adj = 1.0
    temperature *= cooling_rate * adj
    # removed periodic reheating to maintain smoother cooling schedule
    # if (iteration + 1) % (total_iters // 4) == 0:
    # temperature = initial_temperature
    return temperature

def max_min_dis_ratio(n: int, d: int, seed=None):
    """
    Finds n points in d-dimensional space to minimize the max/min distance ratio
```

```
using simulated annealing.

Args:
    n (int): Number of points.
    d (int): Dimensionality of the space.

Returns:
    tuple: (best_points, best_ratio)
"""

# Adaptive hyperparameters based on dimensionality
iterations = 3000 if d <= 2 else 6000 # increased sweeps for improved convergence
initial_temperature = 10.0
cooling_rate = 0.998 if d <= 2 else 0.996 # slower cooling for extended exploration
perturbation_factor = 0.15 if d <= 2 else 0.12 # tuned smaller steps in 3D for better
     local refinement
# relaxation factor for post-acceptance repulsive adjustment
# relaxation_factor removed; using inline 0.1 * perturbation_factor below

# 1. Initial State: reproducible random generator
rng = np.random.default_rng(seed)
# uniform random initialization in [0,1]^d for simplicity
current_points = rng.random((n, d))

_, _, current_ratio = calculate_distances(current_points)

best_points = np.copy(current_points)
best_ratio = current_ratio

temperature = initial_temperature
accept_history = []
window_size = 50 # window for stagnation detection and adaptive injection
# smoothing_interval remains, but smoothing_strength is fixed inlined above
smoothing_interval = max(10, iterations // (20 if d <= 2 else 30)) # more frequent
     smoothing in 3D for improved uniformity

for i in range(iterations):
    # Build KD-tree once per iteration for neighbor queries
    tree = cKDTree(current_points)
    # optional smoothing step using distance-weighted neighbor smoothing
    if (i + 1) % smoothing_interval == 0:
        # choose neighbor count based on dimension
        k_smooth = 6 if d > 2 else 4
        _, idxs = tree.query(current_points, k=k_smooth+1)
        neighbors = current_points[idxs[:,1:]] # exclude self
        # compute inverse-distance weights
        diffs = neighbors - current_points[:, None, :]
        dists = np.linalg.norm(diffs, axis=2) + 1e-6
        weights = 1.0 / dists
        weights /= weights.sum(axis=1, keepdims=True)
        neighbor_means = (neighbors * weights[..., None]).sum(axis=1)
        blend = 0.6 if d > 2 else 0.7
        current_points = np.clip(current_points * blend + neighbor_means * (1 - blend), 0.0,
             1.0)
        _, _, current_ratio = calculate_distances(current_points)
        if current_ratio < best_ratio:
            best_points = current_points.copy()
            best_ratio = current_ratio

    # 2. Generate Neighboring State: Perturb a random point
    # Simplify scaling: rely on temperature to adjust step-size instead of best_ratio
    # dynamic perturbation decays sublinearly with temperature for finer local moves
    perturbation_strength = perturbation_factor * ((temperature / initial_temperature)**0.6
         + 0.15)

    # Choose a random point to perturb
    point_to_perturb_index = rng.integers(0, n)

    old_point = current_points[point_to_perturb_index].copy()
    # Increase repulsivemove frequency in low dimensions
    # dynamic repulsion probability: stronger at high temperature, tapering off as we cool
    if d > 2:
        # reduce repulsion frequency in 3D for finer refinement
        repulsion_prob = float(np.clip(temperature / initial_temperature, 0.2, 0.8))
    else:
        repulsion_prob = float(np.clip(temperature / initial_temperature + 0.1, 0.5, 0.95))
    # start with a random jitter
    # random jitter inlined for readability
    candidate = old_point + rng.uniform(-perturbation_strength, perturbation_strength,
         size=old_point.shape)
    if n > 1 and rng.random() < repulsion_prob:
```

```
            # compute nearest neighbor via KD-tree for efficiency (reusing prebuilt tree)
            _, nn_idxs = tree.query(old_point, k=2)
            nn_idx = nn_idxs[1]
            vec = old_point - current_points[nn_idx]
            norm = np.linalg.norm(vec)
            if norm > 1e-8:
                dir_vec = vec / norm
                candidate = old_point + perturbation_strength * dir_vec
        # keep the point in [0,1]^d
        current_points[point_to_perturb_index] = np.clip(candidate, 0.0, 1.0)
        _, _, candidate_ratio = calculate_distances(current_points)

        # Acceptance criterion
        delta = candidate_ratio - current_ratio
        accept = (delta < 0) or (rng.random() < np.exp(-delta / temperature))

        if accept:
            current_ratio = candidate_ratio
            # Post-acceptance repulsive relaxation to improve local spacing
            # reuse prebuilt KD-tree for repulsive relaxation
            dists, idxs_nn = tree.query(current_points[point_to_perturb_index], k=2)
            dir_vec = current_points[point_to_perturb_index] - current_points[idxs_nn[1]]
            norm = np.linalg.norm(dir_vec)
            if norm > 1e-8:
                # push away from nearest neighbor
                adjustment = 0.1 * perturbation_factor * dir_vec / norm
                current_points[point_to_perturb_index] = np.clip(
                    current_points[point_to_perturb_index] + adjustment, 0.0, 1.0
                )
                # update ratio and best points after relaxation
                _, _, relaxed_ratio = calculate_distances(current_points)
                current_ratio = relaxed_ratio
                if relaxed_ratio < best_ratio:
                    best_points = current_points.copy()
                    best_ratio = relaxed_ratio
            # also keep the standard bestcheck for the candidate move
            if current_ratio < best_ratio:
                best_points = current_points.copy()
                best_ratio = current_ratio
        else:
            current_points[point_to_perturb_index] = old_point

        # Update temperature with adaptive schedule
        accept_history.append(accept)
        temperature = update_temperature(temperature, cooling_rate, accept_history, i,
            iterations, initial_temperature)
        # periodic mild reheating for 3D to escape deep minima
        if d > 2 and (i + 1) % (iterations // 3) == 0:
            temperature = max(temperature, initial_temperature * 0.3)

        # random injection to escape plateaus: reinitialize one point every 20% of iterations
        # random injection only if weve stagnated (low acceptance in recent window)
        if (i + 1) % max(1, iterations // 5) == 0 and len(accept_history) >= window_size \
          and sum(accept_history[-window_size:]) / window_size < 0.1:
            j = rng.integers(0, n)
            current_points[j] = rng.random(d)
            _, _, current_ratio = calculate_distances(current_points)

    # Local refinement stage: fine-tune best solution with small Gaussian perturbations
    refine_iters = max(100, iterations // 20)
    for _ in range(refine_iters):
        idx = rng.integers(0, n)
        old_point = best_points[idx].copy()
        perturb = rng.normal(0, perturbation_factor * 0.05, size=d)
        best_points[idx] = np.clip(old_point + perturb, 0.0, 1.0)
        _, _, refined_ratio = calculate_distances(best_points)
        if refined_ratio < best_ratio:
            best_ratio = refined_ratio
        else:
            best_points[idx] = old_point
    return best_points, best_ratio
```

### I.6 AUTOCONVOLUTION PEAK MINIMIZATION ($L^\infty$).

**Problem Description** For nonnegative densities $f$ supported on $\left[-\frac{1}{2}, \frac{1}{2}\right]$ with $\int f = 1$, define

$$\mu_\infty = \sup_t (f * f)(t).$$

The exact optimum is unknown.

**Human Best:**
$$0.64 \ \le \ \mu_\infty \ \le \ 0.75496.$$

The lower bound is due to Cloninger–Steinerberger, and the upper bound comes from explicit step-function constructions of Matolcsi–Vinuesa (rescaled to unit support).

---

### Initial Proposal

**Problem definition.** Let

$$\mathcal{F} = \left\{ f \in L^1\big([-\tfrac{1}{2}, \tfrac{1}{2}]\big) : \ f \ge 0, \ \int_{-1/2}^{1/2} f(x)\, dx = 1 \right\},$$

and define

$$(f * f)(t) = \int_{\mathbb{R}} f(x)\, f(t - x)\, dx.$$

We seek to minimize the peak value of the autoconvolution:

$$\mu_\infty \ = \ \inf_{f \in \mathcal{F}} \ \|f * f\|_\infty.$$

**Constraints.**
- Nonnegative density.
- Unit mass ($L^1 = 1$).
- Support length 1 (here taken as $[-\tfrac{1}{2}, \tfrac{1}{2}]$).

In the implementation, $f$ is represented by nonnegative step heights on a uniform grid and normalized to unit integral.

**Optimization goal.** Minimize

$$\mu_\infty = \max_t (f * f)(t).$$

Smaller values are better.

**Best-known human results.** In this standard setup, the best currently published bounds are

$$\boxed{0.64 \ \le \ \mu_\infty \ \le \ 0.75496}.$$

The upper bound traces to work of Matolcsi–Vinuesa (after normalizing support length to 1), and the lower bound to Cloninger–Steinerberger.

**Algorithmic goal.** Create an algorithm that constructs feasible densities with progressively smaller $\mu_\infty$. The baseline program generates simple analytical candidates (box, triangle, cosine-squared, Gaussian) on a uniform grid, normalizes to unit mass, and computes autoconvolution via FFT to measure $\mu_\infty$. It serves as a starting point for more advanced search/optimization methods.

**References.**
- E. P. White, *An optimal $L^2$ autoconvolution inequality*, Canadian Mathematical Bulletin (2024).
- M. Matolcsi and C. Vinuesa, *Improved bounds on the supremum of autoconvolutions*, J. Math. Anal. Appl. 372 (2010), 439–447.
- A. Cloninger and S. Steinerberger, *On suprema of autoconvolutions with an application to Sidon sets*, Proc. Amer. Math. Soc. 145 (2017), 3191–3200.

---

**Initial Program:**

```
# -*- coding: utf-8 -*-
"""
Autoconvolution Peak Minimization
=================================
```

```
This program generates step heights for a probability density function
that minimizes the maximum value of its autoconvolution.
"""

import numpy as np
from typing import Dict

def evaluate_C1_upper_std(step_heights: np.ndarray) -> Dict[str, float]:
    """
    Standard-normalized C1 (support [-1/2,1/2], dx=1/K).
    - Project to feasible set: h >= 0 and f = 1 (L1 normalization).
    - Objective: mu_inf = max_t (f*f)(t) (smaller is better).
    Returns: {"valid", "mu_inf", "ratio"(=mu_inf), "integral"(=1.0), "K"}
    """
    h = np.asarray(step_heights, dtype=float)
    if h.size == 0 or np.any(h < 0):
        return {"valid": 0.0, "mu_inf": float("inf"), "ratio": float("inf")}
    K = int(len(h))
    dx = 1.0 / K

    integral = float(np.sum(h) * dx)
    if integral <= 0:
        return {"valid": 0.0, "mu_inf": float("inf"), "ratio": float("inf")}
    h = h / integral # f = 1

    F = np.fft.fft(h, 2*K - 1) # linear autoconvolution via padding
    conv = np.fft.ifft(F * F).real
    conv = np.maximum(conv, 0.0) # clamp tiny negatives

    mu_inf = float(np.max(conv) * dx)
    return {"valid": 1.0, "mu_inf": mu_inf, "ratio": mu_inf, "integral": 1.0, "K": float(K)}

def make_candidate(K: int, kind: str = "cos2") -> np.ndarray:
    """
    Simple candidate builder on [-1/2,1/2] (NOT normalized here).

    Args:
        K: Number of discretization points
        kind: Type of candidate function ("box", "triangle", "cos2", "gauss")

    Returns:
        Step heights array
    """
    x = np.linspace(-1.0, 1.0, K)
    if kind == "box":
        h = np.ones(K)
    elif kind == "triangle":
        h = 1.0 - np.abs(x)
        h[h < 0] = 0.0
    elif kind == "cos2":
        h = np.cos(np.pi * x / 2.0) ** 2
    elif kind == "gauss":
        h = np.exp(-4.0 * x**2)
    else:
        raise ValueError(f"unknown kind={kind}")
    return h

def main():
    """
    Main function that generates step heights for autoconvolution minimization.

    Returns:
        numpy.ndarray: Step heights array
    """
    K = 128
    kind = "cos2" # Change this to try different candidates (box/triangle/cos2/gauss)
    step_heights = make_candidate(K, kind)

    # Evaluate the result to verify it's valid
    result = evaluate_C1_upper_std(step_heights)
    print(f"Generated {kind} candidate with K={K}, mu_inf={result['mu_inf']:.6f}")

    return step_heights
```

## I.7 THIRD AUTOCORRELATION INEQUALITY.

**Problem Description** Let $C_3$ be the largest constant such that $\max_{|t| \leq 1/2} |(f * f)(t)| \geq C_3 \left( \int_{-1/4}^{1/4} f \right)^2$ for all (signed) $f$.

**Best-known:** classical $1.4581$ upper bound.

## I.8 THIRD-ORDER AUTOCORRELATION INEQUALITY ($C_3$ UPPER BOUND)

> **Initial Proposal**
>
> **Problem.**   For piecewise-constant nonnegative functions on a fixed support with unit mass, we evaluate an upper bound $C_{\text{upper\_bound}}$ derived from the maximum of the autoconvolution (normalized by squared $L^1$ mass). The benchmark score is
>
> $$\text{score} = \frac{1}{C_{\text{upper\_bound}}},$$
>
> so that larger score indicates a smaller upper bound and hence a better result.
>
> **Evaluator.**   The evaluator calls `find_better_c3_upper_bound()` from the target program to obtain step heights, computes the normalized autoconvolution maximum, and returns $1/C_{\text{upper\_bound}}$.
>
> **Baseline algorithm.**   A simple genetic algorithm over height sequences serves as the baseline search method. The algorithm includes:
>   - Tournament selection,
>   - One-point crossover,
>   - Gaussian mutation.

**Initial Program:**

```python
import scipy.integrate

def calculate_c3_upper_bound(height_sequence):

    N = len(height_sequence)
    delta_x = 1 / (2 * N)

    def f(x):
        if -0.25 <= x <= 0.25:
            index = int((x - (-0.25)) / delta_x)
            if index == N:
                index -= 1
            return height_sequence[index]
        else:
            return 0.0

    integral_f = np.sum(height_sequence) * delta_x
    integral_sq = integral_f**2

    if integral_sq < 1e-18:
        return 0.0

    t_points = np.linspace(-0.5, 0.5, 2 * N + 1)

    max_conv_val = 0.0
    for t_val in t_points:

        lower_bound = max(-0.25, t_val - 0.25)
        upper_bound = min(0.25, t_val + 0.25)

        if upper_bound <= lower_bound:
            convolution_val = 0.0
        else:
            def integrand(x):
                return f(x) * f(t_val - x)

            convolution_val, _ = scipy.integrate.quad(integrand, lower_bound, upper_bound,
                limit=100)
```

```python
        if abs(convolution_val) > max_conv_val:
            max_conv_val = abs(convolution_val)

    return max_conv_val / integral_sq

def genetic_algorithm(population_size, num_intervals, generations, mutation_rate,
     crossover_rate):

    population = np.random.rand(population_size, num_intervals) * 2 - 1

    best_solution = None
    best_fitness = 0.0

    for gen in range(generations):

        fitness_scores = np.array([calculate_c3_upper_bound(individual) for individual in
            population])

        current_best_idx = np.argmax(fitness_scores)
        if fitness_scores[current_best_idx] > best_fitness:
            best_fitness = fitness_scores[current_best_idx]
            best_solution = population[current_best_idx].copy()
            # print(f"Generation {gen}: New best fitness = {best_fitness}")

        new_population = np.zeros_like(population)
        for i in range(population_size):

            competitors_indices = np.random.choice(population_size, 2, replace=False)
            winner_idx = competitors_indices[np.argmax(fitness_scores[competitors_indices])]
            new_population[i] = population[winner_idx].copy()

        for i in range(0, population_size, 2):
            if np.random.rand() < crossover_rate:
                parent1 = new_population[i]
                parent2 = new_population[i+1]
                crossover_point = np.random.randint(1, num_intervals - 1)
                new_population[i] = np.concatenate((parent1[:crossover_point],
                    parent2[crossover_point:]))
                new_population[i+1] = np.concatenate((parent2[:crossover_point],
                    parent1[crossover_point:]))

        for i in range(population_size):
            if np.random.rand() < mutation_rate:
                mutation_point = np.random.randint(num_intervals)
                new_population[i, mutation_point] += np.random.normal(0, 0.1)

                new_population[i, mutation_point] = np.clip(new_population[i, mutation_point],
                    -2, 2)

        population = new_population

    return best_solution

def find_better_c3_upper_bound():

    NUM_INTERVALS = 4
    POPULATION_SIZE = 2
    GENERATIONS = 10
    MUTATION_RATE = 0.1
    CROSSOVER_RATE = 0.8

    height_sequence_3 = genetic_algorithm(POPULATION_SIZE, NUM_INTERVALS, GENERATIONS,
        MUTATION_RATE, CROSSOVER_RATE)

    return height_sequence_3
```

