# OpenReview forum: "AlphaResearch: Accelerating New Algorithm Discovery with Language Models"
_ICLR.cc/2026/Conference — Submitted to ICLR 2026_

### Official Review · Reviewer_orst · 2025-10-21

**Soundness:** 1
**Presentation:** 2
**Contribution:** 2
**Rating:** 2
**Confidence:** 4

**Summary:**

This paper introduces AlphaResearch, an LLM-based Agent system. Unlike past works that solve problems with pre-defined human answers (e.g., HLE, GAIA), this system attempts to tackle sub-fields that humans have not yet explored. AlphaResearch employs a two-stage process for algorithm discovery, which includes simulated verification with a Reward Model and real-world experimentation. The authors claim to have built an evaluation benchmark named AlphaResearchComp, upon which they conducted a detailed assessment.

I have carefully examined the details of this paper and reviewed related works such as AlphaEvolve, OpenEvolve, and ShinkaEvolve.

**Strengths:**

The paper is the first to introduce a two-stage verification process. It begins by training a Reward Model on real ICLR peer-review data. This model learns to simulate the preferences of real-world researchers, evaluating whether "new ideas" generated by the LLM possess sufficient novelty and feasibility. I appreciate this approach and suggest the authors emphasize its purpose more in the introduction (the current version may leave readers wondering why two verification stages are necessary). I believe this method can reduce the extra resources consumed in the second stage, for instance, by pre-emptively filtering out clearly erroneous examples to better allocate the costs of algorithm discovery.

**Weaknesses:**

The core weakness of this paper is that its contribution is similar to a series of past works, making its improvements incremental.

1.  To my knowledge, AlphaEvolve also employed the iterative process mentioned in line 14: "(1) propose new ideas; (2) program to verify; (3) optimize the research proposals." Therefore, this cannot be considered a core contribution of this paper. As far as I can tell, only the "reward from simulated real-world peer review environment" is a novel point in this work, but the authors fail to emphasize it clearly, instead using vague language that conflates their work with AlphaEvolve.

    For example, line 045 states: "AlphaEvolve (Novikov et al., 2025) introduces an evolutionary coding agent that could tackle open scientific problems with program-based verification. However, the absence of real-world research environment rewards in coding-only agents (Tian et al., 2024) renders the discovery of out-of-boundary knowledge and algorithms challenging for current autonomous research agents."

    This sentence is clearly misleading. I do not believe the "However" can be used to describe AlphaEvolve, as it did incorporate rewards from a real-world research environment. In fact, it is the work by Tian et al. (2024) that lacked real-world rewards.

2.  The paper builds the AlphaResearchComp benchmark, but [OpenEvolve](https://www.google.com/search?q=https://github.com/codelion/openevolve/tree/main/examples) already includes dozens of different algorithm discovery environments. I believe this new benchmark offers little value, as most of its tasks are already covered in OpenEvolve's public examples.

3.  The paper lacks a sufficient discussion and analysis of AlphaEvolve. I did not see a direct comparison, with the only mention appearing in line 313. I am unaware of the specific experimental settings, including time, cost, and number of iterations, but simply citing results from the original paper is inadequate. The original AlphaEvolve paper used the Gemini-2.0 model, whereas AlphaResearch uses o4-mini. I cannot discern the novelty or performance advantages of AlphaResearch from this. In the "n=26 Packing Circles" task, AlphaEvolve achieved 2.635 while AlphaResearch achieved 2.636; this improvement is so marginal that I suspect it could be due to randomness.

4.  The paper is missing a comparison with a range of state-of-the-art baselines, such as OpenEvolve (a codebase that reproduces AlphaEvolve). I would expect the authors to provide adequate comparative experiments to substantiate their core claims.

5.  The use of AlphaResearch-RM-7B is not well-justified. The tasks in AlphaResearchComp are mostly mathematical algorithms, with their human SoTA dating back 10-20 years. The paper provides insufficient evidence to support that a model trained on ICLR data (which mostly covers machine learning and neural networks) can make reasonable judgments on mathematical algorithm tasks. I even suspect that overfitting could be an issue, potentially making the fine-tuned RM less accurate on these math tasks than an untrained LLM.

**Questions:**

The paper is also missing an experiment on cost versus performance. I am curious what the difference would be compared to AlphaResearch if the first-stage RM judgment was omitted. For instance, given the same amount of time or the same budget (API call fees), would AlphaResearch consistently outperform the version without the RM?

---

> ### Author Response · Authors · 2025-11-22
> **Author Response to Reviewer orst (1/3)**
>
> > W1. To my knowledge, AlphaEvolve also employed the iterative process mentioned in line 14: "(1) propose new ideas; (2) program to verify; (3) optimize the research proposals." Therefore, this cannot be considered a core contribution of this paper. As far as I can tell, only the "reward from simulated real-world peer review environment" is a novel point in this work, but the authors fail to emphasize it clearly, instead using vague language that conflates their work with AlphaEvolve.
> For example, line 045 states: "AlphaEvolve (Novikov et al., 2025) introduces an evolutionary coding agent that could tackle open scientific problems with program-based verification. However, the absence of real-world research environment rewards in coding-only agents (Tian et al., 2024) renders the discovery of out-of-boundary knowledge and algorithms challenging for current autonomous research agents."
> This sentence is clearly misleading. I do not believe the "However" can be used to describe AlphaEvolve, as it did incorporate rewards from a real-world research environment. In fact, it is the work by Tian et al. (2024) that lacked real-world rewards.
>
> We appreciate the reviewer’s concern. We’d like to emphasize that the principal difference between AlphaResearch and AlphaEvolve[1] (or ShinkaEvolve[2]) is the different way to get reward from the environment. AlphaEvolve is a coding agent (old program -> new program)  that receives a single reward signal derived solely from program-execution outcomes. AlphaResearch is a research-based agent (old research idea -> new research idea) that leverages two complementary reward sources: (1) program-execution feedback and (2) a simulated peer-review environment.
>
> While the execution-only approach proves effective for optimization within established paradigms, it may create an inherent limitation: the agent can only explore directions that immediately manifest as improved program performance.
> AlphaResearch addresses this constraint by introducing a second reward signal from a simulated peer-review environment trained on OpenReview data. The intuition here mirrors scientific research—breakthrough ideas often require exploring paths that open new conceptual territories without necessarily executing them. A peer-review reward model captures qualities that execution metrics miss: theoretical soundness, methodological rigor, and potential for broader impact.
> This dual-reward structure enables AlphaResearch to explore and refine research directions beyond the capabilities of coding-only agents.
>
> We have updated the experiments  in Appendix A and believe the confusion can be readily addressed by polishing the intro as we already updated in the new PDF.
>
>
> > W2. The paper builds the AlphaResearchComp benchmark, but OpenEvolve already includes dozens of different algorithm discovery environments. I believe this new benchmark offers little value, as most of its tasks are already covered in OpenEvolve's public examples.
>
> The value of AlphaResearchComp is not only the problem set itself. but its standardized and fully documented construction pipeline. While OpenEvolve offers many environments, it functions mainly as a codebase: task definitions, initialization choices, constraint handling, and evaluation rules are embedded implicitly in code and are not specified as benchmark standards. This makes it difficult to ensure reproducibility or conduct fair comparisons across agents.
>
> In contrast, AlphaResearchComp provides explicit, academically defined problem formulations, verification rules, and unified metrics (e.g., excel@best), enabling reproducible and controlled evaluation for open-ended discovery. This standardized pipeline design is essential for studying research agents and cannot be achieved by directly relying on OpenEvolve’s implementation. We have already made this distinction clear in the revision.
>
> Additionally, as we depicted in Section 3, we collect 4 inherited problems from prior work (e.g., AlphaEvolve) and 4 unique problems collected from online repositories and domain experts.
> | AlphaResearch                                     | OpenEvolve                 |
> |---------------------------------------------------|----------------------------------|
> | packing circles (n=26)                            | circles_packing                               |
> | packing circles (n=32)                            | circles_packing                                |
> | minimizing max-min distance ratio (d=2, n=16)     | minimizing_max_min_dist          |
> | third autocorrelation inequality                  | third_autocorr_ineq              |
> | spherical code (n=30)                             | -                                   |
> | autoconvolution peak minimization (upper bound)  | -                                |
> | littlewood polynomials (n=512)                    | -                                |
> | MSTD (n=30)                                      | -                                           |

---

> ### Author Response · Authors · 2025-11-22
> **Author Response to Reviewer orst (2/3)**
>
> > W3. The paper lacks a sufficient discussion and analysis of AlphaEvolve. I did not see a direct comparison, with the only mention appearing in line 313. I am unaware of the specific experimental settings, including time, cost, and number of iterations, but simply citing results from the original paper is inadequate. The original AlphaEvolve paper used the Gemini-2.0 model, whereas AlphaResearch uses o4-mini. I cannot discern the novelty or performance advantages of AlphaResearch from this. In the "n=26 Packing Circles" task, AlphaEvolve achieved 2.635 while AlphaResearch achieved 2.636; this improvement is so marginal that I suspect it could be due to randomness.
>
> We have updated the experiments of the comparison with OpenEvolve (o4-mini) in **Appendix A** of our new submissions.
> For open-ended discovery problems, the absolute number is not a good metric for the evaluation. For example, on "n=26 Packing Circles" task,  AlphaEvolve achieved 2.635
> while the human best results achieved 2.634. The improvement on absolute number of AlphaResearch and AlphaEvolve is both approximately 0.001. So from the perspective of absolute numbers, it's difficult to conclude that this improvement is so marginal.
> This is also why we introduced the new excel@best metric in the AlphaResearchComp benchmark.
>
> In addition, a direct comparison with AlphaEvolve is not feasible because its reported results depend on a different model backbone (Gemini-2.0) and undocumented environment settings, making controlled head-to-head evaluation impossible. Our comparisons therefore use OpenEvolve, which is the open-source reproduction of AlphaResearch.
>
> > W4. The paper is missing a comparison with a range of state-of-the-art baselines, such as OpenEvolve (a codebase that reproduces AlphaEvolve). I would expect the authors to provide adequate comparative experiments to substantiate their core claims.
>
> Thank you for the comment. We have added the experiment results comparing with OpenEvolve on the other six problems in Figure 7 of Appendix A. Our key findings are as follows:
>
> - First 4 problems: AlphaResearch demonstrates a more efficient discovery process than OpenEvolve (open source version of AlphaEvolve), which shows the effectiveness of our dual environments for research-based agents.
>
> - Last 2 problems (littlewood polynomials, MSTD): Both AlphaResearch and OpenEvolve fail to improve the "out-of-the-shelf" algorithm performance, which reveals the limitation of current long-horizon agents where they are not able to explore the search space efficiently on out-of-the-shelf solutions.
>
>
> For the comparison with ShinkaEvlove [1], we have updated it in **Appendix D** of the new PDF.

---

> ### Author Response · Authors · 2025-11-22
> **Author Response to Reviewer orst (3/3)**
>
> > W5. The use of AlphaResearch-RM-7B is not well-justified. The tasks in AlphaResearchComp are mostly mathematical algorithms, with their human SoTA dating back 10-20 years. The paper provides insufficient evidence to support that a model trained on ICLR data (which mostly covers machine learning and neural networks) can make reasonable judgments on mathematical algorithm tasks. I even suspect that overfitting could be an issue, potentially making the fine-tuned RM less accurate on these math tasks than an untrained LLM.
>
> We acknowledge that training the RM on ICLR data for mathematical algorithmic tasks may seem counterintuitive. Rather than claiming perfect transfer, we focus on what our experiments demonstrate.
> Empirically, we find that AlphaResearch-RM-7B consistently improves exploration efficiency without compromising correctness. Across all tasks in AlphaResearchComp, the RM-guided agent never underperforms the baseline without RM (see Appendix A for detailed comparisons). This holds even though final correctness is always determined by program-based verification, independent of the RM's judgments.
>
> Why might this work despite the domain mismatch? We hypothesize that the RM may be capturing general research methodology, scientific claim verification, logical coherence and methodological rigor, and clear articulation of novelty. It’s possible that even an imperfect reward signal can help prune poor research directions early in the search process.
>
> For now, our focus is end-to-end analysis and our results suggest that even a potentially domain-mismatched RM provides practical benefits when combined with rigorous program-based verification, functioning as a useful exploration guide. There are clear future extensions to our work such as training domain-specific reward models and analyzing which features/skills exactly transfer across domains.
>
> Regarding overfitting: if the RM had overfit to ML-specific patterns at the expense of general reasoning, we would expect it to make poor judgments on mathematical tasks, potentially worse than an untrained LLM as the reviewer suggests. Yet empirically, we observe the opposite: the RM consistently helps rather than hinders exploration. This suggests the fine-tuning process preserved or enhanced its ability to evaluate research quality rather than degrading it through overfitting. We refer to the new experiments in Appendix A.
>
> > Q1. The paper is also missing an experiment on cost versus performance. I am curious what the difference would be compared to AlphaResearch if the first-stage RM judgment was omitted. For instance, given the same amount of time or the same budget (API call fees), would AlphaResearch consistently outperform the version without the RM?
>
> Thank you for your suggestions. We have added the experiment cost analysis in Appendix B.
>
> To directly address your question, Figure 7 in Appendix A now comparesAlphaResearch with OpenEvolve  (open source version of AlphaEvolve where RM judgement was omitted) on 6/8 problems of AlphaResearchComp. AlphaResearch demonstrates a more efficient discovery process than OpenEvolve on the first 4 tasks and remains the same on the last 2, which shows the effectiveness of our dual environments in accelerating open-ended discovery..
>
> [1] AlphaEvolve: A coding agent for scientific and algorithmic discovery. (Novikov et al.,)
> url: https://arxiv.org/pdf/2506.13131
>
> [2] ShinkaEvolve: Towards Open-Ended And Sample-Efficient Program Evolution (Lange et al.,)
> url: https://arxiv.org/pdf/2509.19349

---

> > ### Comment · Reviewer_orst · 2025-11-24
> >
> > Dear Authors,
> >
> > To start with the conclusion: I have decided to **maintain my score**.
> >
> > I have re-read the revised version of your paper, the concerns raised by other reviewers (including your responses to them), and your specific response to me. Setting aside any issues I might have overlooked during my initial review and focusing solely on the 5 points I previously raised, here are the reasons why I am maintaining my score:
> >
> > 1.  Regarding the first point, I remain unconvinced by your logic. In lines 049–053 of the new version, I see no supporting evidence to prove your claim. For example, why would the optimization results achieved by AlphaEvolve in "packing circles" and "spherical code" lack scientific value? (In my view, since performance was improved, they should possess at least some degree of scientific value). The statements you have made here are too arbitrary.
> >
> > 2.  On the other hand, I find it difficult to understand why AlphaResearcher is uniquely capable of producing scientifically valuable results. Even assuming the Reward Model (RM) can filter for "scientific value" with perfect accuracy, the system still relies strongly on the performance of the generative model. For AlphaEvolve: If it generates 100 ideas and 5 have scientific value, you find those 5 after 100 verifications. For AlphaResearcher: Because of the RM, you might only need 10 verifications to find those 5 ideas (assuming 100% recall). However, in principle, it is hard to see *why* this generates scientifically valuable ideas that AlphaEvolve cannot, as the ideas generated at the initial stage should be identical. AlphaResearcher merely adds RM filtering, which seemingly functions only as an accelerator.
> >
> > 3.  Regarding OpenEvolve, If you examine the OpenEvolve code, you will find that its examples and documentation also provide clearly defined, academically formulated problem statements, verification rules, and unified metrics. You cannot ignore the contributions of the entire OpenEvolve community (which involves dozens of contributors) simply because it was not written as a formal paper. To do so is disrespectful to the academic community.
> >
> > 4.  Regarding the RM Domain Mismatch, Training `AlphaResearch-RM-7B` on public peer review data from the Machine Learning domain and using it to evaluate ideas for a series of Mathematics tasks still seems unreasonable to me. Reviewers tm64 and SUAh also noted this. I cannot agree with the following statement:
> >
> >     > "For now, our focus is end-to-end analysis and our results suggest that even a potentially domain-mismatched RM provides practical benefits when combined with rigorous program-based verification, functioning as a useful exploration guide."
> >
> >     You should not use one hypothesis to verify or falsify another; this approach lacks rigor.
> >
> > 5.  Regarding Contribution Clarity, Given that you previously blurred the distinction between the contributions of AlphaEvolve and AlphaResearcher—and noting that only Reviewer Vxqg and I recognized this while other reviewers missed it—I believe it is highly necessary to conduct another round of review rather than accepting the paper at this stage.
> >
> > ---
> >
> > Typo: In line 043, please correct **AlphaEvolve \citet{xxx}** to **AlphaEvolve \citep{xxx}**.

---

> > > ### Author Response · Authors · 2025-11-26
> > >
> > > Dear reviewer orst,
> > >
> > > Thank you for your efforts on our paper. We agree with some of your points but would like to clarify the distinction among the contributions of AlphaEvolve, OpenEvolve and AlphaResearch:
> > >
> > > 1. We totally acknowledge the pioneering significance and scientific value of results achieved by AlphaEvolve.
> > > AlphaEvolve can discover novel and valuable algorithms with program revolution.
> > > In lines 049–053 of the new version, we don't claim that the results of AlphaEvolve lacks scientfic value. we use the sentence "but this verification alone might not be completely sufficient for discovery. For example..." to point at the potential problems of the discovery **process** (not the results) of AlphaEvolve, where many code diffs are generated to update the program. This discovery pattern of code diffs could make the process not trackable and advisable enough for human expert. Our solution is adding research content and feedback to the proragm-based process.
> > > We would not like to replace AlphaEvolve with AlphaResearch.
> > > As our paper title "Accelerating new algorithm discovery..." claimed, we hope AlphaReseach could accelerate this discovery process and make the discovery process more trackable and advisable by adding research content and environment to the program-based discovery process in AlphaEvolve. Hence, we agree your comment "AlphaResearcher functions as an accelerator". This aligns to our paper title "Accelerating new algorithm discovery. with LMs".
> > >
> > > 2. We totally acknowledge and respect the achievement of openevolve and the contributors on evolutionary algorithms and application. OpenEvolve provides a great implementation for the public. In previous content, we did not overlook OpenEvolve's contributions. We would not like to replace the problems set of OpenEvolve with AlphaResearchComp. In the data section of our work, we also try to scale the number and diversity of problems with reproducible standard run OpenEvolve on these problems to verify our data collection pipeline.
> > >
> > > We kindly request the reviewer to re-evaluate the contribution of our work and look forward to your further reply.
> > >
> > > Best,
> > >
> > > Authors

---

### Official Review · Reviewer_SUAh · 2025-10-30

**Soundness:** 2
**Presentation:** 3
**Contribution:** 2
**Rating:** 4
**Confidence:** 4

**Summary:**

This paper introduces AlphaResearch, an autonomous agent designed to discover novel algorithms beyond the frontier of human knowledge using Large Language Models (LLMs). The core methodological innovation is a **dual-reward environment** that aims to balance **innovation** by using a reward model (AlphaResearch-RM-7B) trained on real-world peer reviews, and **feasibility** through a rigorous, program-based execution environment. To evaluate this framework, the authors have constructed AlphaResearchComp, a benchmark suite of 8 open-ended algorithmic problems. The experimental results show that AlphaResearch outperforms human researchers on 2/8 problems, notably achieving a new state-of-the-art (SOTA) result on the **Packing Circles (n=32)** problem, surpassing both human records and the prior SOTA agent, AlphaEvolve.

I affirm the importance of the research direction automated scientific discovery pursued in this paper, and find the proposed dual-reward architecture to be sound and well-motivated. However, the paper suffers from numerous shortcomings in its experimental analysis and comparative evaluation. The demonstrated performance gains are limited, suggesting that the work requires further refinement and revision.

**Strengths:**

1. The proposed **dual-reward environment** is a significant and novel contribution. It directly addresses a fundamental challenge in automated scientific discovery: how to balance the pursuit of high-innovation value (proxied by simulated peer review) with the necessity of practical feasibility (ensured by program execution). This decoupled, sequential filtering framework is highly insightful.

2. In NP-hard combinatorial optimization problems like **Packing Circles**, surpassing the SOTA is relatively difficult. The fact that AlphaResearch managed to improve upon a well-established human record and the prior SOTA agent (AlphaEvolve) is a powerful proof-of-concept that clearly demonstrates the potential of the proposed method.

3. The construction of AlphaResearchComp, with its diverse set of problems and varied initialization strategies (e.g., from scratch, improving upon SOTA), is a valuable resource for the emerging field of automated algorithm discovery. It facilitates a more comprehensive assessment of an agent's capabilities across different research scenarios.

**Weaknesses:**

1. A success rate of 2/8 (25%) suggests that the effectiveness of the proposed method may be limited and lacks strong generalizability. It is particularly concerning that the agent failed to make any improvements on the two tasks (**Littlewood polynomials** and **MSTD**) that started from the human-best solution. This strongly implies that AlphaResearch may lack the capability for fine-grained search and optimization within already highly optimized solution spaces.

2. The ablation study reveals that the RM erroneously rejected 43 **viable** ideas, a false negative rate of 28.5% (43/151). For a system aimed at exploring the unknown frontier, prematurely discarding potentially viable paths at such a high rate is a critical risk.

3. This is a core flaw in the study.The RM was trained on ICLR paper abstracts and their review scores, which represent **completed** and **validated** research. It is then applied to evaluate raw, unimplemented, and unrefined **ideas**. Critically, the ideas present in the training data (ICLR abstracts) are, by their nature, ones that have already been proven to be feasible and implementable. This creates a fundamental evaluative bias when the model is used to predict whether a newly generated idea can be successfully implemented by the agent.

4. The paper equates **execution failure** with a **bad idea** and uses this to validate the RM's effectiveness. This is a significant methodological flaw. An execution failure could very well stem from the **limited capabilities of the LLM chosen for code implementation**.

5. The paper's analysis of its experiments and parameters is insufficient. The number of iterations is a key hyperparameter, yet the paper's analysis is superficial. Figure 2 qualitatively shows gains over iterations, but the main results table (Table 4) fails to report the **specific number of iterations** or the **computational cost** required to reach the best solution for each of the 8 tasks.

6. The paper fails to provide a sensitivity analysis for the RM's scoring threshold. How was this threshold determined? How would a more lenient or stringent threshold impact final performance and convergence speed?

7. The paper does not compare its single-trajectory sampling mechanism for idea generation against AlphaEvolve's evolutionary approach. It is therefore impossible to know if the SOTA improvement comes from the dual-reward system or simply from a different search strategy.

I want to reiterate that I agree the research direction of this paper is highly important, and the proposed dual-reward architecture is logical and insightful.

However, the current paper is lacking in its empirical evaluation. The demonstrated performance advantage is not overwhelmingly persuasive, and more critically, the experimental analysis is severely insufficient regarding its core components, key hyperparameters, and agent behavior. This makes it difficult to form a complete understanding of the paper's contributions and limitations. Therefore, I recommend **Major Revision** to address the critical issues raised in the experimental evaluation, ablation studies, and methodological rigor.

**Questions:**

1.  Why did the authors choose **04-mini**, a model with potentially limited coding capabilities, for the critical task of code implementation? How can the authors be certain that the 108 **execution failures** filtered by the RM were due to flawed ideas rather than flawed implementations? If a more powerful code-generation LLM (e.g., GPT-5, as cited in the paper) were used as the implementation agent, would the RM's **correct rejection rate** (71.5%) drop significantly?

2.  How do the authors interpret the RM's high false negative rate of 28.5% (rejecting 43 viable ideas)? Is this an acceptable trade-off between efficiency and innovation? Was a qualitative analysis performed on these 43 erroneously rejected ideas? Do they systematically represent a class of high-risk, high-reward, but **non-ICLR-sounding** innovative paths?

3.  Could the authors provide the number of iterations and the approximate computational cost required to achieve the final SOTA results on the two successful tasks? Furthermore, when the agent was stuck in plateaus on the six unsuccessful tasks, what were the characteristics of the ideas it subsequently generated?

4.  AlphaEvolve uses a (quasi-)evolutionary algorithm to maintain population diversity, whereas AlphaResearch's single-trajectory sampling seems more prone to getting stuck in local optima. Have the authors considered combining their dual-reward framework with a more robust population-based evolutionary strategy (e.g., an island model) to more rigorously test the directive benefit of the RM?

---

> ### Author Response · Authors · 2025-11-22
> **Author Response to Reviewer SUAh (1/3)**
>
> > W1. A success rate of 2/8 (25%) suggests that the effectiveness of the proposed method may be limited and lacks strong generalizability. It is particularly concerning that the agent failed to make any improvements on the two tasks (Littlewood polynomials and MSTD) that started from the human-best solution. This strongly implies that AlphaResearch may lack the capability for fine-grained search and optimization within already highly optimized solution spaces.
>
> We acknowledge the 25% success rate appears limited, but this should be contextualized: previous methods achieve 0% on this benchmark, making our 2/8 a significant advance in autonomous algorithm discovery.
> Although AlphaResearch succeeds on 2 of 8 tasks, these results should be interpreted in light of the structure of the benchmark. The two tasks that begin from human-best solutions (Littlewood polynomials and MSTD) require extremely fine-grained, local improvements on highly rugged landscapes—an area where current long-horizon LLM agents, regardless of search strategy, struggle to make measurable progress.
>
> As shown in the updated Figure 7 in Appendix A, both AlphaResearch and the coding-only OpenEvolve baseline fail on these tasks, indicating that the limitation arises from the precision required by the problem setting. This indicates that the problems are not introduced by AlphaResearch but one limitation of the current long-horizon agents, rather than from the dual-reward framework itself. In contrast, on  other tasks where the search landscape supports conceptual exploration, AlphaResearch consistently discovers improved algorithms, demonstrating that the proposed dual-reward system is effective even though the overall success rate is bounded by the inherent difficulty of the benchmark.
>
>
> > W2. The ablation study reveals that the RM erroneously rejected 43 viable ideas, a false negative rate of 28.5% (43/151). For a system aimed at exploring the unknown frontier, prematurely discarding potentially viable paths at such a high rate is a critical risk.
>
> Although the RM exhibits a 28.5% false-negative rate, we find that most rejected ideas are minor or low-impact variants, and the uniform resampling strategy ensures these ideas are not permanently discarded. We added a case analysis of the discovery process in **Appendix E**.
>
> As shown in Appendix E, the revised draft 4f4c7847 in the rejected pair from checkpoint 634 is effectively identical to its parent e436c26a. Notably, this is found in the later period of the discovery process (Round 632-633). Aside from inflating GA hyperparameters (e.g., population = 300 → 500, generations = 40 → 120) and adding an optional differential_evolution branch, the entire pipeline above find_better_c3_upper_bound is byte-for-byte the same. Crucially, the core loop still calls the undefined normalize_population, triggering the same NameError before any new logic can run. Because this “revision” neither fixes the blocking bug nor implements the promised multi-phase CMA-ES/surrogate/SOS pipeline, it constitutes only a cosmetic variant rather than a substantive new direction.
>
> The RM acts only as a prioritizer to reduce bad directions of research instead of a correctness oracle, and the execution environment ultimately validates feasibility. In aggregate, this dual-reward structure significantly improves exploration efficiency and leads to better overall performance than the coding-only baseline, despite the presence of false negatives. Additionally, as shown in table 2, the prediction accuracy of human annotation is 65.0% (lower than RM's 72.0%), and frontier LLMs like GPT-5 performance are also lower than AlphaResearch-RM, which indicates the difficulty of accurately predicting the success of a research idea and the effectiveness of our RM.

---

> ### Author Response · Authors · 2025-11-22
> **Author Response to Reviewer SUAh (2/3)**
>
> > W3. This is a core flaw in the study.The RM was trained on ICLR paper abstracts and their review scores, which represent completed and validated research. It is then applied to evaluate raw, unimplemented, and unrefined ideas. Critically, the ideas present in the training data (ICLR abstracts) are, by their nature, ones that have already been proven to be feasible and implementable. This creates a fundamental evaluative bias when the model is used to predict whether a newly generated idea can be successfully implemented by the agent.
>
> Thank you for pointing out this issue. Our RM is not used to decide whether an idea is implementable; it only provides a lightweight plausibility prior. AlphaResearch operates in a dual-environment loop: the RM first filters obviously low-quality ideas based on abstract-level coherence, and the execution environment then performs the actual feasibility and performance verification. Only execution results determine whether an idea is valid or surpasses the current best. Because RM scores never serve as the final decision signal, the potential bias from training on validated ICLR abstracts does not propagate into the discovery process. Empirically, this dual-stage design reduces ineffective trials while preserving correctness, indicating that the RM’s high-level heuristics generalize sufficiently without introducing harmful bias.
>
>
> > W4. The paper equates execution failure with a bad idea and uses this to validate the RM's effectiveness. This is a significant methodological flaw. An execution failure could very well stem from the limited capabilities of the LLM chosen for code implementation.
>
> We'd like to clarify that the execution failure of AlphaResearch does not necessarily mean that the idea is a bad one. Those failed executions will be given a 0 execution score and put back into the candidate pool, where they might be selected again in future iterations.  Our view is that promising ideas often require multiple rounds of refinement and polishing. For this reason, we rely on a uniform sampling strategy rather than a greedy selection approach.
>
>
> > W5. The paper's analysis of its experiments and parameters is insufficient. The number of iterations is a key hyperparameter, yet the paper's analysis is superficial. Figure 2 qualitatively shows gains over iterations, but the main results table (Table 4) fails to report the specific number of iterations or the computational cost required to reach the best solution for each of the 8 tasks.
>
> Thank you for your suggestions.
> We have added the experiment parameters (e.g., iterations,computational cost) required to reach the best solution for each of 8 tasks of AlphaResearchComp in Appendix B.
>
> > W6. The paper fails to provide a sensitivity analysis for the RM's scoring threshold. How was this threshold determined? How would a more lenient or stringent threshold impact final performance and convergence speed?
>
> Because the RM was trained on real ICLR peer-review records, its threshold carries practical meaning. We set the threshold to 5.5,  which corresponds to the midpoint between the historical ICLR guidelines for **borderline reject (5) and accept (6)** decisions prior to 2025.This choice mirrors the decision boundary used in actual review practice and allows the training process to more faithfully approximate a realistic peer-review environment.
>
> Conceptually, the threshold only affects prioritization—not correctness. No idea is ever permanently discarded: our uniform resampling ensures rejected candidates re-enter the pool, and correctness is always determined solely by program-based verification.
>
> > W7. The paper does not compare its single-trajectory sampling mechanism for idea generation against AlphaEvolve's evolutionary approach. It is therefore impossible to know if the SOTA improvement comes from the dual-reward system or simply from a different search strategy.
>
> We have updated the experiments of the comparison with OpenEvolve in **Appendix A** of our new submissions. Among the 6/8 failed tasks of AlphaResearch, AlphaResearch surpasses OpenEvolve on 4 tasks, which demonstrate the effectiveness of the dual-reward framework.
>
> But on the Littlewood polynomial and MSTD task, both AlphaEvolve and AlphaResearch fail to make any progress. This reveals the limitation of current long-horizon agents where they **could not be able** to explore the search space efficiently on out-of-the-shelf solutions.

---

> ### Author Response · Authors · 2025-11-22
> **Author Response to Reviewer SUAh (3/3)**
>
> > Q1. Why did the authors choose 04-mini, a model with potentially limited coding capabilities, for the critical task of code implementation? How can the authors be certain that the 108 execution failures filtered by the RM were due to flawed ideas rather than flawed implementations? If a more powerful code-generation LLM (e.g., GPT-5, as cited in the paper) were used as the implementation agent, would the RM's correct rejection rate (71.5%) drop significantly?
>
> We have added the comparison with GPT-5 in **Figure 8 of Appendix C**. As shown in figure 8, AlphaResearch (GPT-5) reaches high performance significantly faster than o4-mini in the early stages of discovery. However, in the later stages, the two models perform comparably, which suggests that their underlying capabilities are close to algorithm discovery tasks.
>
>
> > Q2. How do the authors interpret the RM's high false negative rate of 28.5% (rejecting 43 viable ideas)? Is this an acceptable trade-off between efficiency and innovation? Was a qualitative analysis performed on these 43 erroneously rejected ideas? Do they systematically represent a class of high-risk, high-reward, but non-ICLR-sounding innovative paths?
>
>
> Although the RM shows a 28.5% false-negative rate, this pattern largely reflects the inherent selectivity of the ICLR peer-review signals it is trained on. Importantly, these false negatives do **not** hinder overall discovery: our uniform resampling mechanism ensures rejected ideas are not permanently excluded, and verification is always determined by program-execution rather than the RM alone. As shown in Section W2, the dual-reward system still yields higher final effectiveness than the coding-only baseline, indicating that the RM’s generalizable evaluation heuristics improve exploration efficiency even when some viable ideas are initially filtered out. These results suggest that the trade-off remains acceptable, and that peer-review-derived signals generalize sufficiently to guide search without suppressing innovation.
>
>
> > Q3. Could the authors provide the number of iterations and the approximate computational cost required to achieve the final SOTA results on the two successful tasks? Furthermore, when the agent was stuck in plateaus on the six unsuccessful tasks, what were the characteristics of the ideas it subsequently generated?
>
> In Appendix B,  we have updated the experiment parameters (iterations,computational cost) required to reach the best solution for each of 8 tasks of AlphaResearchComp.
> We also added deeper analysis to the six unsuccessful tasks (comparison with OpenEvolve) in Appendix A.
> On failed tasks, the agent tends to produce minor perturbations of prior ideas rather than conceptual shifts, which explains the observed plateaus.
>
> > Q4. AlphaEvolve uses a (quasi-)evolutionary algorithm to maintain population diversity, whereas AlphaResearch's single-trajectory sampling seems more prone to getting stuck in local optima. Have the authors considered combining their dual-reward framework with a more robust population-based evolutionary strategy (e.g., an island model) to more rigorously test the directive benefit of the RM?
>
> Thank you for your insightful feedback. The success of AlphaEvolve has shown the advantages of population-based evolutionary strategy.
> We believe combining the dual-reward framework of AlphaResearch with other advanced methods (like population-based evolutionary strategy) could promote more exciting discovery. We add the comparison of AlphaResearch with OpenEvolve[1] (open source AlphaEvolve[2]) in **Figure 7** to justify the effectiveness of the dual environment of AlphaResearch, and will discuss any type of possible trials in future research.
>
> [1]https://github.com/algorithmicsuperintelligence/openevolve
>
> [2] AlphaEvolve: A coding agent for scientific and algorithmic discovery. (Novikov et al.,)
> url: https://arxiv.org/pdf/2506.13131

---

### Official Review · Reviewer_tm64 · 2025-11-03

**Soundness:** 3
**Presentation:** 3
**Contribution:** 3
**Rating:** 6
**Confidence:** 5

**Summary:**

This paper proposes a method for training a reward model to efficiently filter ideas during algorithm discovery. It also introduces a new benchmark for algorithm discovery, AlphaResearchComp. AlphaResearch is an agent that utilizes this reward model. The execution reward is used for performance evaluation, while the reward model is employed for idea filtering. This reward is applied immediately after idea generation, and if the score falls below a threshold, that round is skipped. The results on AlphaResearchComp show that AlphaResearch surpasses human researchers on the packing circles problem, while the other tasks remain challenging, demonstrating the current limitations of autonomous algorithm discovery.

**Strengths:**

- Training a reward model specifically for algorithm discovery is a very good idea. There is a gap between reward and generation, and existing methods have centered on program execution rewards.

**Weaknesses:**

- Although the sampling procedure in AlphaResearch is defined as drawing (i_t, p_t, r_t) from the past trajectory (\tau_{k-1}) according to a probability distribution (P(\cdot \mid \tau_{k-1})), the paper does not explain the concrete properties or design principles of that distribution (P). As a result, it is unclear by what criteria past steps are selected, for example uniform selection, recency preference, or performance weighting. The main process of the algorithm, namely new idea generation, depends on this sampling, yet its details are not provided, which reduces clarity.
- If reviews are not publicly available, the RM cannot be trained. In many fields outside machine learning, peer reviews are generally not public, which makes the RM effectively untrainable and limits the method in other domains.
- The RM incurs inference cost. How does it compare to mechanical filtering methods for ideas, such as detecting keyword overlap with past ideas or computing textual similarity to past ideas?
- AlphaResearchComp includes only 6 evaluated problems, which is a rather limited number for drawing general conclusions about the agent’s capabilities.

**Questions:**

- Proposing a new benchmark is an excellent contribution. However, it would be beneficial to include comparisons with existing evaluation frameworks such as PaperBench [1] and MLE-bench [2] in the related work section.
- The RM improves the efficiency of algorithmic idea generation, but is it better than directly training a policy from past ideas [3]?
- The research agent is only o4-mini [4]. Since the performance of discovered algorithms in algorithm discovery depends heavily on the agent’s capability, running experiments with other agents and demonstrating consistency would make the paper more convincing.

[1] PaperBench: Evaluating AI's Ability to Replicate AI Research

[2] MLE-bench: Evaluating Machine Learning Agents on Machine Learning Engineering

[3] Can Large Language Models Invent Algorithms to Improve Themselves?: Algorithm Discovery for Recursive Self-Improvement through Reinforcement Learning

[4] OpenAI o3 and o4-mini System Card

---

> ### Author Response · Authors · 2025-11-22
> **Author Response to Reviewer tm64 (1/2)**
>
> > W1. Although the sampling procedure in AlphaResearch is defined as drawing (i_t, p_t, r_t) from the past trajectory (\tau_{k-1}) according to a probability distribution (P(\cdot \mid \tau_{k-1})), the paper does not explain the concrete properties or design principles of that distribution (P). As a result, it is unclear by what criteria past steps are selected, for example uniform selection, recency preference, or performance weighting. The main process of the algorithm, namely new idea generation, depends on this sampling, yet its details are not provided, which reduces clarity.
>
> Thanks for the insightful comments. We have updated the Method section to provide a better understanding of how AlphaResearch works. We use uniform sampling for the trajectory sampling to ensure that all ideas have the same chance to be refined and executed, thereby avoiding suboptimal algorithm results. This prevents early high-scoring-but-suboptimal ideas from dominating
>
> > W2. If reviews are not publicly available, the RM cannot be trained. In many fields outside machine learning, peer reviews are generally not public, which makes the RM effectively untrainable and limits the method in other domains.
>
> Thank you for your insightful feedback.  The real-world research environments are highly dynamic and complex. One solution of the challenge is to collect more high-quality peer review environments (or data). For other domains where peer-reviewed is not public,  the lack of good peer review environments make it challenging to train a model that can make reasonable judgments on these tasks. Therefore, extending the RM’s ability to generalize on unseen domains is an important direction for future research.
> Our work, AlphaResearch, shows that training RM models using ICLR records may also improve their performance on mathematical algorithm discovery tasks and contributes to construct a fair and realistic research environment for LLMs.
>
>
>
> > W3. The RM incurs inference cost. How does it compare to mechanical filtering methods for ideas, such as detecting keyword overlap with past ideas or computing textual similarity to past ideas?
>
> Thanks for your feedback. We'd like to clarify that what AlphaResearch contributes is to construct a fair and realistic research environment for language models just like what top ML conferences like ICLR does for human researchers.
> Mechanical filtering methods like keyword overlap indicate the internal features of model generated content and are hard which could not give the agent effective real world reward externally. But it could be a great topic for future work because these
> internal features could measure other mechanics in research agents like agent memory.
>
> > W4. AlphaResearchComp includes only 6 evaluated problems, which is a rather limited number for drawing general conclusions about the agent’s capabilities.
>
> We agree on the importance of scaling the number of test problems.The collection of open-ended algorithm discovery problems is relatively a  challenging task that needs a lot of manual effort. We plan to release a larger dataset in the future.
>
> From the 8 problems of AlphaResearchComp, we can obtain 3 conclusions for the topic of "accelerating human-level algorithm discovery with language models":
>
> 1. Among the 8 problems, AlphaResearch outperforms the results of human researcher and AlphaEvolve[1](Coding-only) on packing circles (n=26, n=32) problems, which demonstrate the potential of accelerating human-level algorithm discovery with language models.
>
> 2. We add the experiment results of the other 6 problems in **Figure 7 of Appendix A**. AlphaResearch demonstrates a more efficient discovery process than OpenEvolve (open source version of AlphaEvolve) on the first 4 tasks, which shows the effectiveness of our dual environments for research-based agents.
>
> 3. On last 2 problems (littlewood polynomials, MSTD), Both AlphaResearch and OpenEvolve fail to improve the "out-of-the-shelf" algorithm performance, which reveals the limitation of current long-horizon agents where they are not able to explore the search space efficiently on out-of-the-shelf solutions.
>
> [1] AlphaEvolve: A coding agent for scientific and algorithmic discovery. (Novikov et al.,)
> url: https://arxiv.org/pdf/2506.13131

---

> ### Author Response · Authors · 2025-11-22
> **Author Response to Reviewer tm64 (2/2)**
>
> > Q1. Proposing a new benchmark is an excellent contribution. However, it would be beneficial to include comparisons with existing evaluation frameworks such as PaperBench [1] and MLE-bench [2] in the related work section.
>
> Thank you for your suggestions. We have updated our **related work** section to include these great works!
>
>
> > Q2. The RM improves the efficiency of algorithmic idea generation, but is it better than directly training a policy from past ideas [3]?
>
> Reinforcement learning is also a good way to justify the effectiveness of our dual environment.In this paper, we mainly talk about the training-free self-evolving ability with the proposed research-based dual environment where the reward is encoding in the prompt of the research agent to promote further algorithm discovery. The discussion about rl training will be included for future research.
>
>
> > Q3.The research agent is only o4-mini [4]. Since the performance of discovered algorithms in algorithm discovery depends heavily on the agent’s capability, running experiments with other agents and demonstrating consistency would make the paper more convincing.
>
> We have added the comparison with GPT-5 in **Figure 8 of Appendix C**. As shown in figure 8, AlphaResearch (GPT-5) reaches high performance significantly faster than o4-mini in the early stages of discovery. However, in the later stages, the two models perform comparably, which suggests that their underlying capabilities are close on algorithm discovery task.

---

### Official Review · Reviewer_Vxqg · 2025-11-03

**Soundness:** 2
**Presentation:** 2
**Contribution:** 2
**Rating:** 4
**Confidence:** 4

**Summary:**

This paper introduces AlphaResearch, an autonomous research agent designed to discover novel algorithms that surpass human-created ones. Unlike prior coding-only systems, AlphaResearch combines LLM-based idea generation, program implementation, and optimization within a dual environment: a simulated peer-review system (AlphaResearch-RM-7B) and program-based verification. Through iterative ideation, coding, and evaluation, AlphaResearch generates, tests, and refines research ideas autonomously. In experiments on eight open-ended problems, AlphaResearch outperforms human researchers on two tasks and achieves state-of-the-art results beyond AlphaEvolve. The study demonstrates LLMs’ potential to advance human knowledge while identifying remaining challenges in autonomous algorithm discovery.

**Strengths:**

The paper tackles an ambitious and novel problem—whether LLM-based agents can autonomously discover algorithms that go beyond human knowledge boundaries. The design of a dual environment that combines peer-review-style reward modeling with executable program verification represents a meaningful conceptual advance over prior coding-only or LLM-as-a-judge approaches. If validated, the proposed framework could represent an important step toward autonomous scientific discovery, a key open question in AI research. The demonstration that AlphaResearch outperforms human experts on certain tasks, even if limited, provides intriguing preliminary evidence of LLM potential beyond human knowledge boundaries.

**Weaknesses:**

1. While combining peer-review simulation with execution-based verification is an interesting integration, the overall conceptual novelty is modest. The work mainly repackages existing ideas from prior frameworks such as AlphaEvolve and LLM-as-a-judge.
2. The description of the AlphaResearch pipeline lacks sufficient procedural clarity. It is unclear how the system refines research ideas, translates them into executable code, and performs iterative optimization. The explanation of interactions between modules (LLM, RM, executor) is particularly vague.
3. The experimental setup is under-specified and potentially insufficient to substantiate the paper’s claims. Key issues include (a) limited number of test problems (only 8), (b) unclear evaluation metrics, (c) missing baseline comparisons, and (d) lack of dataset details.
4. The comparisons with existing approaches (e.g., coding-only or LLM-as-a-judge systems) are superficial. The paper does not adequately isolate which component—peer review or execution-based verification—contributes most to the performance gains.
5. The paper claims that AlphaResearch “pushes the boundary of human knowledge,” but the results do not convincingly support this. Success on two problems out of eight, without robust statistical validation or expert human evaluation, limits the credibility of such claims.

**Questions:**

LN127: what is the contribution of the paper? Besides the proposed AlphaResearch and AlphaResearchCamp, what are your findings or conclusion from your research? Need to describe it explicitly.

LN161: The method description seems incomplete. Are you trying to formulate the method in the framework of RL? If it is true, the peer-review and execution provides reward signals, and a policy is expected to be optimized using these signals. However, the policy and optimization algorithm are not provided in the section.

LN192: The experimental results and discussion should go to Experiments section.

LN217: What is the unit for Human Best and what is the metric?

LN260: The definition of excel@best is problematic. Considering I_d=1 and a fixed r_human, a r_best=r_human-1 and a r_best=r_human+1 could result in the same excel@best.

LN306: Your approach for evaluation, including datasets, baselines, and metrics, are expected here.

LN310: Are 8 problems sufficient to reach a solid conclusion? Need a justification in the settings section.

LN360: The comparison seems unfair given that AlphaResearch filters the ideas based on the evaluation provided by AlphaResearch-RM-7B. Additionally, figure 3 shows that AlphaResearch is better. However, it is hard to quantify the differences from the figure itself.

LN454: What is advantage of the combination compared to existing studies? Maybe you can justify it here.

LN485: It is hard to make conclusion based on the results and observations.

---

> ### Author Response · Authors · 2025-11-22
> **Author Response to Reviewer Vxqg (1/3)**
>
> > W1. While combining peer-review simulation with execution-based verification is an interesting integration, the overall conceptual novelty is modest. The work mainly repackages existing ideas from prior frameworks such as AlphaEvolve and LLM-as-a-judge.
>
> AlphaResearch introduces a fundamentally different reward mechanism from prior work. In comparison, AlphaEvolve is a coding agent that only a single execution-based reward (old program → new program), whileAlphaResearch is a research-based agent (old research idea -> new research idea) that could obtain dual reward from the program execution and simulated peer review environment. This dual-reward design is the core novelty of our framework and is what enables AlphaResearch to guide research exploration more effectively than coding-only or judge-only systems.
>
>
> > W2. The description of the AlphaResearch pipeline lacks sufficient procedural clarity. It is unclear how the system refines research ideas, translates them into executable code, and performs iterative optimization. The explanation of interactions between modules (LLM, RM, executor) is particularly vague.
>
> The AlphaResearch agent iteratively: (1) generate novel ideas and filter candidate novel ideas with **AlphaResearch-RM**  (2) the **LLM** converts a selected idea into executable code, and (3) the **executor** runs the generated program and provides performance feedback for the next refinement step. This interaction between the RM, LLM, and executor enables the agent to avoid unpromising directions and refine ideas efficiently over iterations.
>
>
>
>
> > W3. The experimental setup is under-specified and potentially insufficient to substantiate the paper’s claims. Key issues include (a) limited number of test problems (only 8), (b) unclear evaluation metrics, (c) missing baseline comparisons, and (d) lack of dataset details.
>
>
> W3.(a) We agree on the importance of scaling the number of test problems. The collection of open-ended algorithm discovery problems is relatively a challenging task that needs a lot of manual effort. Please note that the nature of experiments make them expensive to run and therefore, baselines such as ShinkaEvolve [1] also have similar scope of experiments (During the discovery, each problem takes ~2-5 days to evaluate)
> Both the verification of current human best and the design of the evaluation code takes a lot of time (can be weeks).We plan to release a larger dataset in the future.
>
> W3.(b) For open-ended discovery problems, we notice that the absolute number is not a good metric for the evaluation. For example, on "n=26 Packing Circles" task,  AlphaEvolve[1] achieved 2.635
> while the human best results achieved 2.634. The improvement on absolute number of AlphaResearch and AlphaEvolve is both approximately 0.001. So from the perspective of absolute numbers, it's difficult to conclude that this improvement is so marginal. Therefore, we design the excel@best metric to evaluate the percentage of improvement over the human best results. On "n=26 Packing Circles" task, the 0.001 improvement of AlphaResearch corresponds to a 0.32 percent gain over the human best result. We will clarify this in the new submission.
>
> W3.(c) We have updated and analyzed the experiments of the comparison with OpenEvolve (o4-mini),the open sourced version of AlphaEvolve, in the **Appendix A** of our new submissions.
>
> W3.(d) We publish the dataset details of AlphaResearchComp in the **Appendix G**  of our new submissions, including the definition of the problems, the algorithm goal and the initial algorithms.  All these open-ended discovery problems will be released publicly in the future.
>
> > W4. The comparisons with existing approaches (e.g., coding-only or LLM-as-a-judge systems) are superficial. The paper does not adequately isolate which component—peer review or execution-based verification—contributes most to the performance gains.
>
> **Comparison with Coding-only agent** Thank you for your suggestions. We updated the experiments of the comparison with OpenEvolve (coding-only) in Appendix A of our new submissions. Among the 6 selected  tasks of AlphaResearch, AlphaResearch surpasses OpenEvolve on 4 tasks, which demonstrate the effectiveness of the dual-reward framework.
> But on the Littlewood polynomial and MSTD task, both AlphaEvolve and AlphaResearch fail to make any progress. This reveals the limitation of current long-horizon agents where they are not able to explore the search space efficiently on out-of-the-shelf solutions.
>
> **Compare with agent without judger(RM)** As shown in Figure 5, we compare the performance of AlphaResearch with and without the judger (AlphaResearch-RM-7B). The judger (AlphaResearch-RM-7B) helps save the error rate of idea execution from 51.5% to 24.5%, which greatly helps AlphaResearch to avoid ineffective trials and discover feasible research ideas more efficiently.

---

> ### Author Response · Authors · 2025-11-22
> **Author Response to Reviewer Vxqg (2/3)**
>
> > W5. The paper claims that AlphaResearch “pushes the boundary of human knowledge,” but the results do not convincingly support this. Success on two problems out of eight, without robust statistical validation or expert human evaluation, limits the credibility of such claims.
>
> Thank you for your suggestions. We have revised the phrasing of this conclusion, so it better reflects the contribution of our work: Our revised framing positions AlphaResearch as advancing autonomous algorithm discovery through dual-reward architectures, rather than claiming to expand human knowledge.
>
>
> > Q1. LN127: what is the contribution of the paper? Besides the proposed AlphaResearch and AlphaResearchCamp, what are your findings or conclusions from your research? Need to describe it explicitly.
>
> The core contribution of AlphaResearch is to construct a powerful research agent and a fair and realistic research dual environment for language models and show that the research-based dual environment could help research agents to discover the unknown and accelerate the algorithm discovery.
>
>
> First, we demonstrate that combining program-execution rewards with peer-review rewards enables more effective exploration of research spaces compared to execution-only feedback. Our experiments show that AlphaResearch has the capacity to improve human performance on verifiable algorithm discovery tasks, illustrating the potential of automatic AI scientific discovery processes.
>
> Second, we find that reward models trained on peer-review data can provide useful guidance. Our RM successfully guides mathematical algorithm discovery, suggesting the potential of a peer-review based environment to help LLMs in the scientific discovery process.
>
> Third, our results reveal that current language models, when equipped with appropriate environmental feedback, can autonomously traverse complex research spaces. This represents an advance from previous coding-only agents that optimize existing solutions to agents that can reformulate problems and discover fundamentally different approaches.
>
>
> > Q2. LN161: The method description seems incomplete. Are you trying to formulate the method in the framework of RL? If it is true, the peer-review and execution provides reward signals, and a policy is expected to be optimized using these signals. However, the policy and optimization algorithm are not provided in the section.
>
> In this paper, we mainly talk about the training-free self-evolving ability (not a training method) with the proposed research-based dual environment where the reward is encoding in the prompt of the research agent to promote further algorithm discovery. We did not involve reinforcement learning at current stage, and will discuss it in future research.
>
>
> > Q3. LN192: The experimental results and discussion should go to Experiments section.
>
> Thank you for your suggestions. We have moved the discussion to the Experiments section.
>
>
> > Q4. LN217: What is the unit for Human Best and what is the metric?
>
> **Percentage (%)**. We design the excel@best metric to evaluate the percentage of improvement over the human best results. For example, on "n=26 Packing Circles" task, AlphaResearch achieved 0.001 improvement, which is 0.32% of the human best results.
>
> > Q5. LN260: The definition of excel@best is problematic. Considering I_d=1 and a fixed r_human, a r_best=r_human-1 and a r_best=r_human+1 could result in the same excel@best.
>
> The excel@best metric to evaluate the percentage of improvement over the human best results. Therefore, for I_d=1, if r_best=r_human-1, the excel@best is negative, if r_best=r_human+1, the excel@best is positive. For I_d=-1, if r_best=r_human-1, the excel@best is positive, if r_best=r_human+1, the excel@best is negative. We have adjusted Equation 4 to adopt this.

---

> ### Author Response · Authors · 2025-11-22
> **Author Response to Reviewer Vxqg (3/3)**
>
> > Q6. LN306: Your approach for evaluation, including datasets, baselines, and metrics, are expected here.
>
> Dataset:
>
>
> From the 8 problems of AlphaResearchComp, we collect 3 kinds of problems:
>
> (1) For the “Packing Circles” (n=26) and “Packing Circles” (n=32) problems, we initialize them with null programs (r0 = 0) to simulate research starting from scratch.
>
> (2) For the “Littlewood Polynomials” and “MSTD (n=30)” problems, we directly adopt the best-known solutions (r0 = r_human) from human researchers to emulate improvements upon established methods.
>
> (3) For the remaining problems, we employ a moderate initialization strategy (0 < r_0 < r_human) to ensure sufficient room for the research agent to explore.
>
> Metrics: the performance of the best algorithm during the discovery process.
>
> Baseline:
> We compare AlphaResearch with the evolutionary coding agent OpenEvolve in Appendix  and obtain the following observations:
>
> 1. Among the 8 problems, AlphaResearch outperforms the results of human researcher and AlphaEvolve (Coding-only) on packing circles (n=26, n=32) problems, which demonstrate the potential of accelerating human-level algorithm discovery with language models.
>
> 2. We add the experiment results of the other 6 problems in **Figure 7 of Appendix A**. AlphaResearch demonstrates a more efficient discovery process than OpenEvolve (open source version of AlphaEvolve) on the first 4 tasks, which shows the effectiveness of our dual environments for research-based agents.
>
> 3. On last 2 problems (littlewood polynomials, MSTD), Both AlphaResearch and OpenEvolve fail to improve the "out-of-the-shelf" algorithm performance, which reveals the limitation of current long-horizon agents where they are not able to explore the search space efficiently on out-of-the-shelf solutions.
>
>
>
> > Q7. LN310: Are 8 problems sufficient to reach a solid conclusion? Need a justification in the settings section.
>
> AlphaResearchComp provides **explicit, academically** defined problem formulations, verification rules, and unified metrics (e.g., excel@best), enabling reproducible and controlled evaluation for open-ended discovery. We standardized the 8 problems with clear task definitions, initialization choices, constraint handling, and evaluation rules covering 3 real-world research scenarios. This standardized **pipeline design** is essential for studying research agents and cannot be achieved by directly relying on previous implementations like AlphaEvolve[2].
>
>
> > Q8. LN360: The comparison seems unfair given that AlphaResearch filters the ideas based on the evaluation provided by AlphaResearch-RM-7B. Additionally, figure 3 shows that AlphaResearch is better. However, it is hard to quantify the differences from the figure itself.
>
> To clarify, our comparison does not only compare  “executed ideas” but rather the full distribution of all 1000 generated ideas from both agents. This avoids any bias introduced by filtering. In Figure 3, we report the score distribution over all generated ideas (not only those executed), showing that AlphaResearch produces higher-quality ideas even before filtering.
>
> > Q9. LN454: What is advantage of the combination compared to existing studies? Maybe you can justify it here.
>
> The main advantage of our approach is that peer-review guidance and execution-based verification provide complementary signals: the RM reduces unproductive exploration by filtering low-value ideas, while the execution environment validates feasibility and drives actual improvement. This dual-reward structure enables AlphaResearch to explore more selectively and improve more consistently than existing coding-only or judge-only system
>
> **Comparison with Coding-only agent** Thank you for your suggestions. We updated the experiments of the comparison with OpenEvolve (coding-only) in Appendix A of our new submissions. Among the 6/8 failed tasks of AlphaResearch, AlphaResearch surpasses OpenEvolve on 4 tasks, which demonstrate the effectiveness of the dual-reward framework.
> But on the Littlewood polynomial and MSTD task, both AlphaEvolve and AlphaResearch fail to make any progress. This reveals the limitation of current long-horizon agents where they are not able to explore the search space efficiently on out-of-the-shelf solutions.
>
> **Compare with agent without judger(RM)** As shown in Figure 5, we compare the performance of AlphaResearch with and without the judger (AlphaResearch-RM-7B). The judger (AlphaResearch-RM-7B) helps save the error rate of idea execution from 51.5% to 24.5%, which greatly helps AlphaResearch to avoid ineffective trials and discover feasible research ideas more efficiently.
>
> We have also updated this information in **Appendix A**.
>
> > Q10. LN485: It is hard to make conclusion based on the results and observations.
>
> We have revised the phrasing of this conclusion to better reflect the contribution of our work to the understanding of how research agents could accelerate the discovery of new algorithms.

---

> > ### Author Response · Authors · 2025-11-24
> > **Reference**
> >
> > [1] ShinkaEvolve: Towards Open-Ended And Sample-Efficient Program Evolution (Lange et al.,)
> > url: https://arxiv.org/pdf/2509.19349
> >
> > [2] AlphaEvolve: A coding agent for scientific and algorithmic discovery. (Novikov et al.,)
> > url: https://arxiv.org/pdf/2506.13131

---

### Comment · Area_Chair_H5ae · 2025-11-24
**Discussion Period**

Dear reviewers,

The discussion period is now open. Please use the “Official Comments” to engage in discussions about each other's reviews and the authors' rebuttal, and update your assessments or comments as appropriate.

Did the authors' rebuttal adequately address your concerns? We kindly ask that you update your reviews based on these discussions and your evaluation of the rebuttal, even if your overall assessment remains unchanged.

Thank you all for your contributions.

Best regards, AC

---

### Author Response · Authors · 2025-12-04
**General Response**

Dear ACs/SACs/PCs and reviewers,

Thank you for the thoughtful reviews and constructive suggestions.

## Our contributions

Our work has several important contributions:
- The sota research agent AlphaResearch and its implementation.
- The idea to **combine** execution-based research agent like AlphaEvolve and real-world peer review environment (we acknowledge prior work for each aspect individually, but only combining both can lead to tasks with realistic autonomous research), which we believe are two key gap of today’s research agent.
- The frontier algorithms discovered by AlphaResearch.
- The scaling recipe of **evaluation environments** for research agent in **AlphaResearchComp**.

We added 4 Appendix part for comparing AlphaResearch with sota research agents and helping understand AlphaResearch better.

- Comparing AlphaResearch with OpenEvolve and ShinkaEvolve in Appendix A and D. AlphaResearch provides better process score towards more advanced algorithm (Reviewer Vxqg, tm64, SUAh, orst).

- Cost of AlphaResearch in Appendix B (Reviewer SUAh, orst)
| Problem                                          | Iterations | Cost per iteration ($) |
|--------------------------------------------------|------------|-----------------------------|
| packing circles (n=26)                           | 4768       | 0.013                       |
| packing circles (n=32)                           | 4768       | 0.013                       |
| minimizing max-min distance ratio (d=2, n=16)    | 4400       | 0.017                       |
| third autocorrelation inequality                 | 1366       | 0.012                       |
| autocorrelation peak minimization (upper bound)  | 979        | 0.013                       |
| littlewood polynomials (n=512)                   | 2233       | 0.011                       |
| MSTD (n=30)                                      | 2826       | 0.011                       |
| spherical code                                   | 1132       | 0.015                       |
- Comparison with different LLM backbone that illustrates good generalizability of AlphaResearch in Appendix C. (Reviewer SUAh, tm64)
- More detailed trajectory analysis in Appendix E. (Reviewer SUAh)

**In summary, for today’s  autonomous research agent, which are small-scale and relatively simple, we find that the simulation of real-world research environments could help them for improvement over human best.** However, if future tasks evolve to be large-scale and significantly more complex, we’d expect today’s data and naive methods for environments scaling to be less tractable for evaluation. In such a future case, we believe it is important to study better ways to construct and evaluate sota research agent. We will revise our draft and incorporate such a discussion.

Best,

Authors

---

### Meta-Review · Area_Chair_xis1 · 2026-01-01

**Summary:**

The major weaknesses that the reviewers mention are, first, the scale of the test benchmark set is not sufficient to prove that AlphaResearch is helpful (tm64, Vxqg). Even more importantly, the successful test problems where AlphaResearch performs better than humans are even fewer (2) (SUAh).

Second, there is a fundamental flaw in using an ICLR reviews-trained model as RM, as it has only tested positive ideas, without negatives (Vxqg); moreover, knowledge from them is not really applicable beyond ideas in the ML domain (SUAh, orst, tm64).

Third, many reviewers agree that the novelty is fairly incremental on top of AlphaEvolve (orst, Vxqg) and discounts the work done by OpenEvolve, both in methodology and benchmark.

**Reviewer Concerns:**

The authors have done a great job of both improving the writing and distinguishing it from prior work; moreover, the new comparisons to prior work are incredibly useful. However, many of the weaknesses remain outstanding.

Both the benchmark and the benchmark where the models succeed are too small for empirical validation of AlphaResearch. If the authors believe OpenEvolve's benchmark is not up to the mark, maybe a future step could be to make the problems in those more standardized. Secondly, in order to prove effectiveness on a variety of problems and not just circle packing, the authors need to show that AlphaResearch discovers new things not in prior human knowledge beyond circle packing.

Training the RM model on ICLR reviews and testing on Math problems is a fundamental flaw, and the authors have not been able to convince the reviewers or the AC as to why this is the correct setup. In the revision, the authors should either figure out a way to get reviews from the mathematical domain, or in the test benchmark include ML problems/setups.

Reviewer orst is still not convinced of the technical novelty of this work/benchmark over openevolve and I'm inclined to agree with them.

**Reviewer Scores:**

All the reviewers are likely to keep their scores.

---

### Decision · Program_Chairs · 2026-01-26

Reject